# Decoding mechanism of action and sensitivity to drug candidates from integrated transcriptome and chromatin state

Caterina Carraro[1]\*, Lorenzo Bonaguro[2,3]\*, Jonas Schulte-Schrepping[2,3], Arik Horne[2,3], Marie Oestreich[2], Stefanie Warnat-Herresthal[2,3], Tim Helbing[4], Michele De Franco[1], Kristian Haendler[2,5,6], Sach Mukherjee[7,8], Thomas Ulas[2,3,5], Valentina Gandin[1], Richard Goettlich[4], Anna C Aschenbrenner[2,3,5,9], Joachim L Schultze[2,3,5], Barbara Gatto[1]

[1]Department of Pharmaceutical and Pharmacological Sciences, University of Padova, Padova, Italy; [2]Systems Medicine, Deutsches Zentrum für Neurodegenerative Erkrankungen (DZNE) e.V., Bonn, Germany; [3]Genomics and Immunoregulation, Life & Medical Sciences (LIMES) Institute, University of Bonn, Bonn, Germany; [4]Institute of Organic Chemistry, Justus Liebig University Giessen, Giessen, Germany; [5]PRECISE Platform for Genomics and Epigenomics, Deutsches Zentrum für Neurodegenerative Erkrankungen (DZNE) e.V. and University of Bonn, Bonn, Germany; [6]Institute of Human Genetics, University of Lübeck, Lübeck, Germany; [7]Statistics and Machine Learning, Deutsches Zentrum für Neurodegenerative Erkrankungen (DZNE) e.V., Bonn, Germany; [8]MRC Biostatistics Unit, University of Cambridge, Cambridge, United Kingdom; [9]Department of Internal Medicine and Radboud Center for Infectious Diseases (RCI), Radboud University Medical Center, Nijmegen, Netherlands

**\*For correspondence:**
caterina.carraro.2@phd.unipd.
it (CC);
lorenzobonaguro@uni-bonn.de
(LB)

**Competing interest:** The authors declare that no competing interests exist.

**Abstract** Omics-based technologies are driving major advances in precision medicine, but efforts are still required to consolidate their use in drug discovery. In this work, we exemplify the use of multi-omics to support the development of 3-chloropiperidines, a new class of candidate anticancer agents. Combined analyses of transcriptome and chromatin accessibility elucidated the mechanisms underlying sensitivity to test agents. Furthermore, we implemented a new versatile strategy for the integration of RNA- and ATAC-seq (Assay for Transposase-Accessible Chromatin) data, able to accelerate and extend the standalone analyses of distinct omic layers. This platform guided the construction of a perturbation-informed basal signature predicting cancer cell lines' sensitivity and to further direct compound development against specific tumor types. Overall, this approach offers a scalable pipeline to support the early phases of drug discovery, understanding of mechanisms, and potentially inform the positioning of therapeutics in the clinic.

## Editor's evaluation

To test differential anticancer drug effects on different tissue types, and to understand drug response mechanism, the authors set up a series of RNA-seq and ATAC-seq experiments on drug responsive and non-responsive cell lines. Then they conducted bioinformatic analyses to pinpoint networks that are altered in responsive vs non-responsive cell lines. Remarkably, they used their analytic results to calculate tumor- and sample-specific response to the drug.

## Introduction

Omic technologies have revolutionized the classical hypothesis-driven paradigm of drug discovery, offering a new perspective for the systematic identification of targets and therapeutics (*Dugger et al., 2018*; *Paananen and Fortino, 2020*). An increasing number of examples describe the use of these approaches to inspect the pharmacological profile of existing drugs, e.g., mechanism of action (MoA) and specific sensitivity biomarkers, as well as to assist their correct repositioning in clinical practice (*Koromina et al., 2019*; *Matthews et al., 2016*; *Mun et al., 2020*; *Tsimberidou, 2015*). Compared to traditional approaches, omics-based methods capture the complexity of biological systems and pathological processes in its entirety at increasingly affordable costs (*Matthews et al., 2016*). Indeed, significant cost reductions have been announced for sequencing technologies, making them accessible to the scientific community with no impact on their robustness (*Pennisi, 2022*). For this reason, refined strategies to handle the high-dimensional information of omics data are continuously investigated to expedite their routine use in preclinical drug development (*Kagohara et al., 2020*; *Li et al., 2014*; *Li et al., 2021*; *McFarland et al., 2020*; *Rendeiro et al., 2020*; *Shaheen et al., 2018*; *Srivatsan et al., 2020*; *Ye et al., 2018*).

Studies from our group highlighted 3-chloropiperidines (3-CePs) as a novel class of candidate anticancer agents developed to improve the pharmacological profile of nitrogen mustard-based chemotherapeutics, characterized by a fast and affordable synthesis (*Carraro et al., 2021*; *Carraro et al., 2019*; *Helbing et al., 2020*; *Sosic et al., 2017*; *Zuravka et al., 2015a*; *Zuravka et al., 2014*; *Zuravka et al., 2015b*). As intended, these agents were demonstrated to induce DNA lesions, a mechanism conceivably responsible for their cytotoxicity on tested cancer cell lines (*Carraro et al., 2021*; *Carraro et al., 2019*; *Helbing et al., 2020*). Interestingly, despite their expected broad-acting MoA, a subset of derivatives showed preferential activity against pancreatic adenocarcinoma (PAAD)BxPC-3 cells in 2D and 3D cell culture when compared to other treatment-resistant cell lines (e.g. HCT-15 from colorectal cancer). This preferential activity requires more investigation for further preclinical and clinical translation, especially in light of the broad resistance of pancreatic tumors to most of the available treatments (*Carraro et al., 2021*; *Carraro et al., 2019*; *Helbing et al., 2020*).

The contribution of multi-omics to support early phases of drug discovery is growing exponentially in the era of precision medicine (*Li et al., 2021*). Combined omics technologies have the potential to address some of the intrinsic difficulties of the traditional drug discovery and development path, assisting it early from target prioritization and hit identification up to the evaluation of candidates' efficacy and safety (*Mun et al., 2020*). Drug-perturbation experiments have been employed to inspect the functionality of target proteins (*Faivre et al., 2020*) and the MoA of therapeutics, efficiently guiding the decision-making process in the development of lead compounds (*Mun et al., 2020*). The massive accumulation of genomic and transcriptomic profiles offers a precious substrate for the optimization of strategies capable of predicting susceptibility to known therapeutics (*Barretina et al., 2012*; *Lamb et al., 2006*; *Subramanian et al., 2017*; *Uhlen et al., 2017*). These approaches are further refined by the continuous acquisition of data from high-throughput single-cell platforms (*Bush et al., 2017*; *Corsello et al., 2020*; *McFarland et al., 2020*; *Srivatsan et al., 2020*; *Ye et al., 2018*). Beyond the most common transcriptome analysis, changes in gene regulation can be evaluated in terms of chromatin accessibility (*Buenrostro et al., 2013*; *Granja et al., 2019*; *Schmidl et al., 2019*). We hypothesized that combining the information from the transcriptome and chromatin state would enable a higher-resolution and extremely robust mapping of cell dynamics upon drug exposure. Examples of the joint use of these two omic techniques exist (*Kagohara et al., 2020*; *Rendeiro et al., 2020*; *Suzuki et al., 2019*; *Tung et al., 2021*), but their synergistic employment on compounds under early development is still underexplored (*Matthews et al., 2016*).

In this study, representative mono- (**M**) and bi-functional (**B**) 3-CePs bearing a single or double alkylating units (*Figure 1A*) were selected to exemplify the use of a multi-omic approach to investigate the molecular determinants of susceptibility to novel drug candidates and their MoA (*Carraro et al., 2021*; *Carraro et al., 2019*; *Helbing et al., 2020*). We analyzed transcriptional changes and chromatin status of cancer cell lines previously identified as high- (PAAD BxPC-3) and low-sensitive (colorectal adenocarcinoma HCT-15) after treatment by RNA- and ATAC-seq. In addition, we implemented our multi-omics pipeline in drug discovery to derive perturbation-informed signatures predicting compound sensitivity. We validated the approach by assessing cancer cell sensitivity to both the early-discovered compound **M** and the established antineoplastic agent cisplatin. Overall,

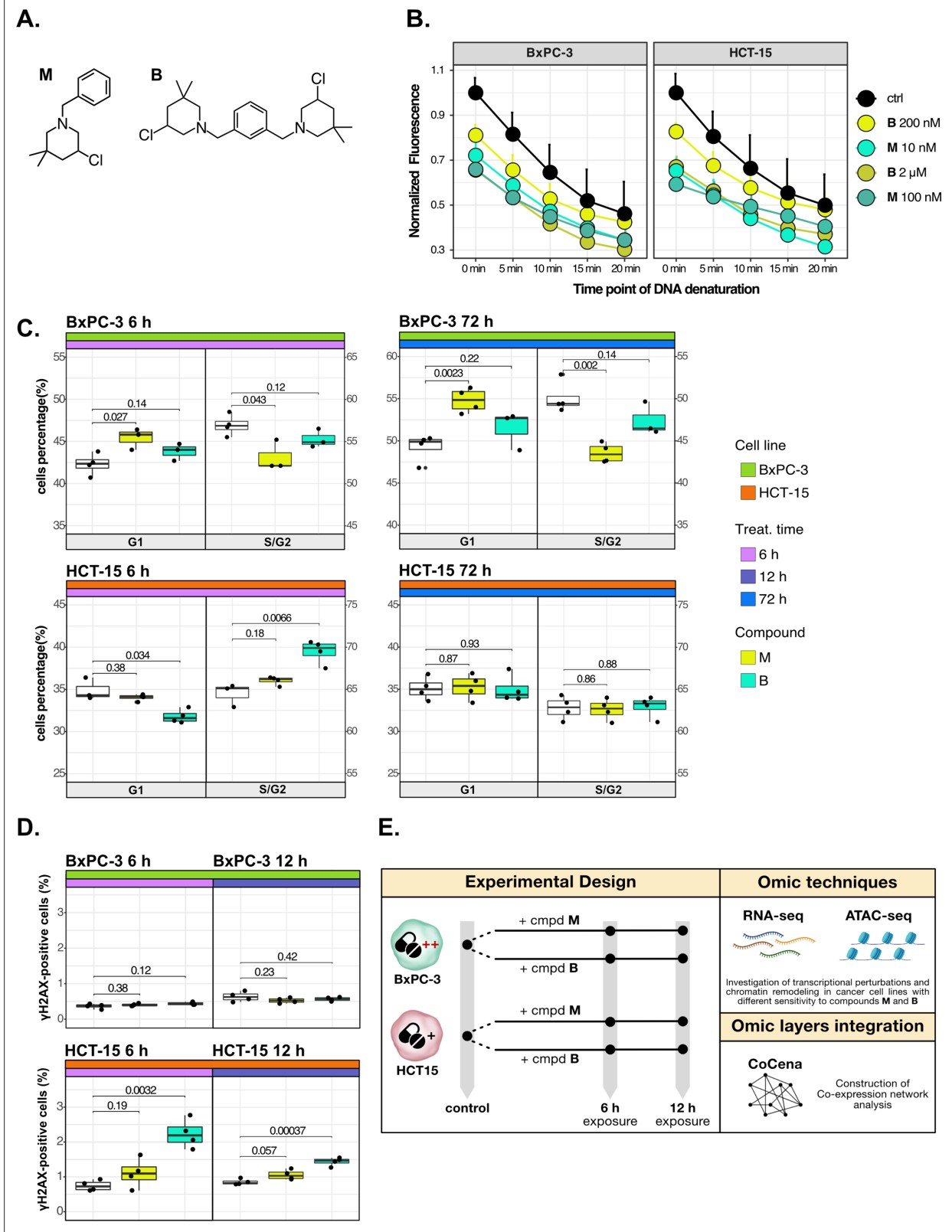

**Figure 1.** Cancer tropism of 3-chloropiperidines (3-CePs) is not explained by DNA damage. (**A**) Chemical structure of the analyzed 3-CePs (M=monofunctional and B=bifunctional). (**B**) Quantification of genomic DNA damage in BxPC-3 and HCT-15 cells treated with M (10 nM and 100 nM), B (200 nM and 2 µM), or DMSO (dimethyl sulfoxide) 0.5% (ctrl) for 6 hr and analyzed by the fast micromethod single-strand-break assay: alkaline denaturation of DNA is followed in time up to 20 min by monitoring the fluorescence of the dsDNA-specific PicoGreen dye. (**C**) Cell cycle distribution

*Figure 1 continued on next page*

Figure 1 continued

(accumulation in G1 vs G2/S phases) of BxPC-3 and HCT-15 cells treated with M (10 nM), B (200 nM), or DMSO 0.5% for 6 hr and 72 hr analyzed by FACS (Fluorescence Activated Cell Sorting). At least three biological replicates were obtained per condition and unpaired two-tailed Student's *t*-test was performed to assess statistical significance (p<0.05). (**D**) Analysis of H2AX phosphorylation in BxPC-3 and HCT-15 cells treated with M (10 nM), B (200 nM), or DMSO 0.5% for 6 hr and 12 hr analyzed by FACS. At least three biological replicates were obtained per condition, and unpaired two-tailed Student's *t*-test was performed to assess statistical significance (p<0.05). (**E**) Schematic representation of the adopted omic-based approach.

The online version of this article includes the following figure supplement(s) for figure 1:

**Figure supplement 1.** Cancer tropism of 3-CePs is not explained by DNA damage—supporting data.

---

the proposed pipeline not only allowed to identify potentially more susceptible target tumor types for further development of test compounds, but also offered a complete predictive framework to support precision oncology in a clinical setting.

## Results

### Cancer tropism of 3-CePs is not explained by DNA damage

The selected **M** and **B** 3-CePs (*Figure 1A*) were shown in previous work to be particularly active in the nanomolar range against BxPC-3 PAAD cells (*Carraro et al., 2019*; *Helbing et al., 2020*). From this premise, the two compounds were chosen along with the highly sensitive BxPC-3 cell line and the low-sensitive HCT-15 colorectal adenocarcinoma cell line (72 hr $IC_{50}$ HCT-15/$IC_{50}$ BxPC-3: 50 for **M**, 10 for **B**) (*Carraro et al., 2019*; *Helbing et al., 2020*) to illustrate how integrative omics approaches unveil the molecular mechanisms responsible for the described cellular tropism.

First, to assess whether 3-CePs-induced DNA damage itself would differ in the two cell lines upon treatment, we measured the accumulation of DNA single-strand breaks after 6 hr of treatment with both compounds at their cytotoxicity $IC_{50}$s in BxPC-3 and at a 10-times higher concentration (10 nM and 100 nM **M**; 200 nM and 2 μM **B**) (*Schröder et al., 2006*). Surprisingly, the two cell lines showed very comparable DNA damage accumulation, in both cases higher after treatment with **M** compared to **B** (*Figure 1B*). These results clearly pointed toward differential responses in the two cell lines downstream of DNA damage.

Since alkylating agents are known to alter the progression of the cell cycle (*Kaufmann and Paules, 1996*; *Meyn and Murray, 1984*; *Tu et al., 2004*), we next performed a cell cycle distribution analysis by flow cytometry after different times of treatment (6 hr, 12 hr, and 72 hr) with both compounds (*Figure 1C*, *Figure 1—figure supplement 1A* and B). While **M** induced a persisting block in G1 throughout the observation time in BxPC-3 cells, this block was absent in HCT-15 cells. In contrast, **B** induced an early G2/S block in HCT-15 cells (6 hr), which was not observed at later time points, while such a block was most obvious at 12 hr for BxPC-3 cells. Despite similar DNA damage accumulation, these findings clearly indicated a different behavior for the two cancer cell lines in terms of cell cycle progression after treatment with the two 3-CePs.

To determine the additional mechanisms explaining differential sensitivity to 3-CePs, we measured the activation of the DNA repair machinery as another key aspect in the cellular response to genotoxicants (*Li et al., 2020*). To verify the ability of the two cancer cell lines to detect double-strand breaks (DSBs), we assessed the phosphorylation of H2AX (γH2AX), an early event of the DNA damage response (DDR) (*Sharma et al., 2012*), by flow cytometry after 6 hr and 12 hr of treatment with both agents (*Figure 1D*). Interestingly, despite the comparable DNA damage accumulation in the two cell lines, only HCT-15 showed an increase in the γH2AX-positive population, suggesting a more efficient engagement of the DNA repair machinery.

Taken together, these results indicated that cell-specific mechanisms after the first event of DNA damage are responsible for the different sensitivity to 3-CePs.

### Treatments elicit cell-specific transcriptional changes

Different genetic and epigenetic factors define the responsiveness of tumor cells to chemotherapeutic agents (*Vasan et al., 2019*). To address these globally, we analyzed changes in the transcriptome of the high- and low-sensitive cell lines after treatment with the two 3-CePs (*Figure 1E*). RNA-seq was performed on total RNA of HCT-15 and BxPC-3 cells exposed to DMSO 0.5% (control) or treated

with **M** (10 nM) or **B** (200 nM) for 6 hr and 12 hr (*Figure 2A*, *Figure 2—figure supplement 1A*) as in previous experiments.

Principal component analysis (PCA) of all transcripts separated samples within each cell line according to treatment and time point (*Figure 2—figure supplement 1B*), suggesting a clear transcriptional reprogramming after treatment. In fact, differential expression (DE) analysis showed that the expression of a large number of genes changed significantly in both cell lines after exposure to 3-CePs ($p < 0.05$, independent hypothesis weighting [IHW] multiple testing correction) (*Figure 2B*, *Figure 2—figure supplement 1C*), especially at 6 hr in BxPC-3 cells and upon treatment with **B** in HCT-15 cells.

Gene ontology (GO) enrichment was performed on the DE genes to determine signaling pathways and transcriptional programs explaining the observed differences. In a first explorative approach, we generated the union of DE genes per cell line irrespective of compound and time point, which allowed us also to distinguish between cell type-specific and shared DE genes (*Figure 2—figure supplement 1D*). The most representative biological processes identified by this analysis (*Figure 2—figure supplement 1E*, *Supplementary file 1*) are reported in *Figure 2C* (see Methods and *Figure 2—figure supplement 1F* for further details).

Unexpectedly, we identified a strong translational response in BxPC-3 cells after treatment, a process which is typically attenuated in stress conditions, as was the exposure to our DNA damaging agents, to allow proper recovery of the protein quality control machinery (*Burger et al., 2010*; *Shenton et al., 2006*). In contrast, a strong regulation of genes mediating protein stability and catabolism was observed in the low-sensitive cell line. In addition, HCT-15 cells activated genes involved in the DDR, consistent with their higher ability to detect and respond to DSBs. Both these two mechanisms pointed toward the activation of an adaptive stress response in the low-sensitive cell line.

To further characterize these transcriptional changes over time in a cell type-specific context, we grouped the DE genes at 6 hr and 12 hr in modules according to the similarity in their expression profiles and performed a functional enrichment on genes with similar expression patterns (*Figure 2D* and *Figure 2—figure supplement 2A*, *Supplementary file 2*). Genes involved in ribosome biogenesis and DNA repair turned out to be upregulated, particularly after 6 hr of treatment in BxPC-3 cells (Clusters 2 and 3, *Figure 2D*). Besides, silencing of pro-survival genes involved in microtubule organization and the JAK-STAT cascade (Cluster 1, *Figure 2D*) was detected at the same time point. Only after 12 hr of treatment (*Figure 2—figure supplement 2A*), BxPC-3 cells boosted carbohydrate metabolism, most likely an attempt to recover in extremis (*Lin et al., 2020*).

Also, HCT-15 cells upregulated clusters of genes mediating DNA repair, protein stability, and mitochondrial activity as early as 6 hr of treatment, suggesting this time point as the most informative to describe the response to 3-CePs (Clusters 4 and 6, *Figure 2D*). In contrast to BxPC-3 cells, HCT-15 downregulated genes involved in translation and ribosome biogenesis from 6 hr of exposure (Cluster 7, *Figure 2D*), while intensifying their response to oxidative stress after 12 hr (Cluster 17, *Figure 2—figure supplement 2A*).

This exploratory analysis showed clearly different transcriptional responses and distinct time dynamics in BxPC-3 compared to HCT-15 cells, most likely responsible for their different susceptibility to 3-CePs. In particular, our findings pointed toward DNA repair and proteostasis as key mechanisms tuning sensitivity to the compounds, as further confirmed by inspecting the complete rank of DE genes via gene set enrichment analysis (GSEA, *Figure 2—figure supplement 2B*; *Subramanian et al., 2005*).

## DNA repair and proteostasis are key modulators of the response to 3-CePs

For their key role in the response to 3-CePs, DNA repair and protein homeostasis were further analyzed to clarify their contribution to BxPC-3 sensitivity.

Interestingly, DNA repair was activated in both cell lines early after 6 hr of treatment but with a different modulation (*Figure 3A*). First, base-excision repair (BER) was suggested as the preferential pathway of BxPC-3 by GO enrichment, while HCT-15 relied mostly on nucleotide-excision repair (NER), unleashing a generally stronger activation of the DDR. In detail, HCT-15 DE genes contributing to the response to the DNA damage stimulus were strongly upregulated already after 6 hr especially in response to **B**, while activated only after 12 hr in BxPC-3 (*Figure 3—figure supplement 1A*). In

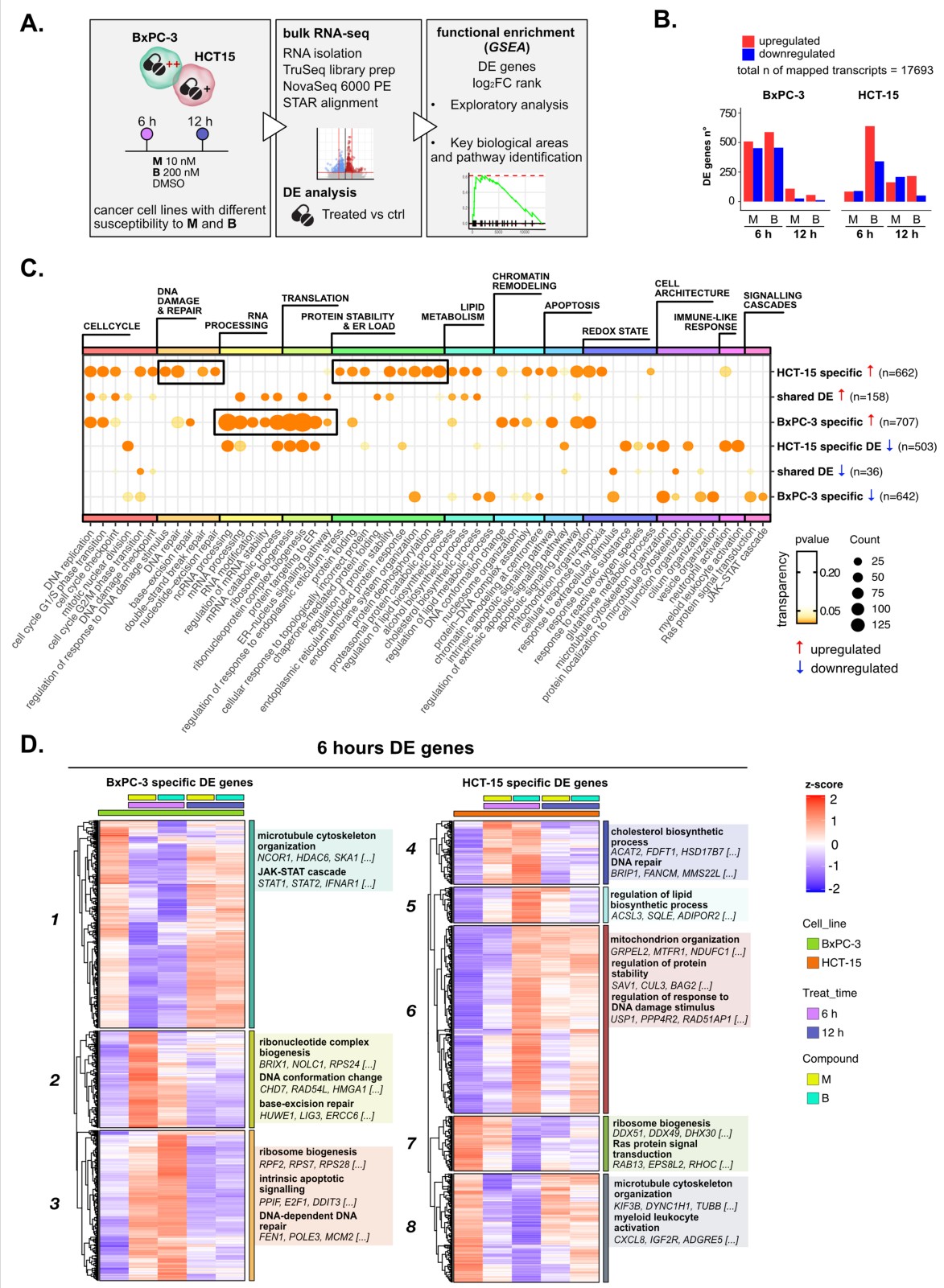

**Figure 2.** Treatments elicit cell-specific transcriptional changes. (**A**) An overview of the applied workflow for the RNA-seq analysis. (**B**) Number of up- (red) and down-regulated (blue) differential expression (DE) genes in BxPC-3 and HCT-15 cells after treatment with M (10 nM), B (200 nM), or DMSO 0.5% (ctrl) for 6 hr and 12 hr (adjusted p threshold = 0.05, shrinkage = TRUE, and multiple testing method = independent hypothesis weighting [IHW]). (**C**) Gene ontology (GO) database functional enrichment gene set enrichment analysis (GSEA) on cell-specific and shared up- and down-regulated DE

*Figure 2 continued on next page*

*Figure 2 continued*

genes. For each identified biological process, enrichments in terms of count and p-value of representative terms are reported (p<0.05). (**D**) Expression level of cell-specific 6 hr DE genes across test conditions. GSEA was performed on modules with similar regulation identified by hierarchical clustering: for each cluster, representative GO terms and genes of the associated load are reported.

The online version of this article includes the following figure supplement(s) for figure 2:

**Figure supplement 1.** Treatments elicit cell-specific transcriptional changes—supporting data.

**Figure supplement 2.** Treatments elicit cell-specific transcriptional changes—supporting data 2.

contrast, genes such as *PPP4R2* and *RAD51AP1*, both involved in the first phases of DSBs repair (*Chowdhury et al., 2008*; *Modesti et al., 2007*), were even downregulated in BxPC-3 cells at 6 hr.

The more efficient activation of DNA repair in HCT-15 was further confirmed on the overall rank of genes by GSEA at 6 hr of treatment (*Figure 3B*). As anticipated, most of the DE genes leading the enrichment in HCT-15 belonged to NER (e.g. *GTF2H3* and *RBX1*) and other recombinational pathways such as homologous repair (HR) (e.g. *MMS22L* and *BARD1*) and fanconi anemia (FA) (e.g. *BRIP1* and *FANCM*), all better suited for the efficient repair of bulky lesions and highly toxic DSBs and crosslinks (*Cantor et al., 2001*; *Li and Jin, 2012*; *Niraj et al., 2019*; *Piwko et al., 2016*; *Satoh and Hanawalt, 1996*; *Westermark et al., 2003*). On the other hand, DE genes in BxPC-3 cells were mostly related to BER (e.g. *APEX1* and *UNG*) and MMR (mismatch repair) (e.g. *MSH6* and *EXO1*), which contribute to the repair of smaller lesions and mismatches (*Fortini et al., 2003*; *Kunkel and Erie, 2005*).

In the analysis, proteostasis was identified as a second key biological process strictly related to genotoxic stress (*González-Quiroz et al., 2020*; *Hutt and Balch, 2010*). HCT-15 cells engaged the protein folding and catabolism apparatus in response to 3-CePs, especially to **B** already at the early time point (*Figure 3C*). As observed for DNA repair, DE genes contributing to protein catabolism were upregulated as early as 6 hr of exposure in HCT-15 cells, whereas they were downregulated at the same time point in BxPC-3 and only upregulated after 12 hr (*Figure 3—figure supplement 1B*). This response involved chaperones and co-chaperones (e.g. *HSPA8*, *HSPA1B*, *BAG2*, and *BAG5*), other genes mediating protein catabolism (e.g. *LAMP2* and *CUL3*), and ER morphogenesis (e.g. *RTN4*) (*Eskelinen, 2006*; *Jozsef et al., 2014*; *Ran et al., 2007*; *Scott et al., 2016*). Interestingly, a transcriptional pattern revealed by GSEA at 6 hr of treatment highlighted an intense positive modulation of the PERK-mediated branch of the unfolded protein response (UPR) specifically in BxPC-3 (*Figure 3D and E*). Even more enlightening were the DE genes leading the enrichment: *ATF4*, *DDIT3* (CHOP), and *PPP1R15A* (GADD34) were significantly upregulated after 6 hr of exposure only in this cell line (*Figure 3F*). These genes participate in the PERK-mediated UPR triggering cell death after prolonged ER stress through the aberrant recovery of translation, which induces proteotoxicity (*Han et al., 2013*; *Urra et al., 2013*). This mechanism would reasonably explain the ribosome biogenesis signature observed in BxPC-3 cells. Consistently, recent work reported a particular susceptibility for pancreatic cancer adenocarcinoma to ER stress and protein dyshomeostasis (*Garcia-Carbonero et al., 2018*).

Furthermore, the ability of HCT-15 cells to control proteostasis may also depend on the activation of lipid and cholesterol biosynthesis in response to the compounds (*Figure 3—figure supplement 1C, D*). In fact, among other known pro-survival functions, these pathways contribute to resolving ER stress through pathways involving, e.g., the Stearoyl-CoA Desaturase (*SCD*) enzyme, for which we detected a significant upregulation of the respective transcript in HCT-15 (*Figure 3—figure supplement 1D*; *Romero et al., 2018*; *Tadros et al., 2017*).

Overall, the transcriptome analysis of this in vitro perturbation experiment allowed us to dissect the different responses to 3-CePs in our model cell lines, pointing toward protein homeostasis and DDR imbalances as mechanisms responsible for the high susceptibility of BxPC-3 cells.

## The response to 3-CePs is further regulated at the chromatin level

The transcriptome analysis unveiled a defined framework of responses tuning the sensitivity to 3-CePs. To further characterize them at the epigenetic level, we examined chromatin accessibility in nuclei of BxPC-3 and HCT-15 cells treated with **M** and **B** for 6 hr and 12 hr (*Figure 4A*, *Figure 4—figure supplement 1A*) by ATAC-seq.

3-CePs induced evident epigenetic changes in both cell lines, as suggested by PCA (*Figure 4—figure supplement 1B*) and confirmed by the number of differentially accessible regions (DARs)

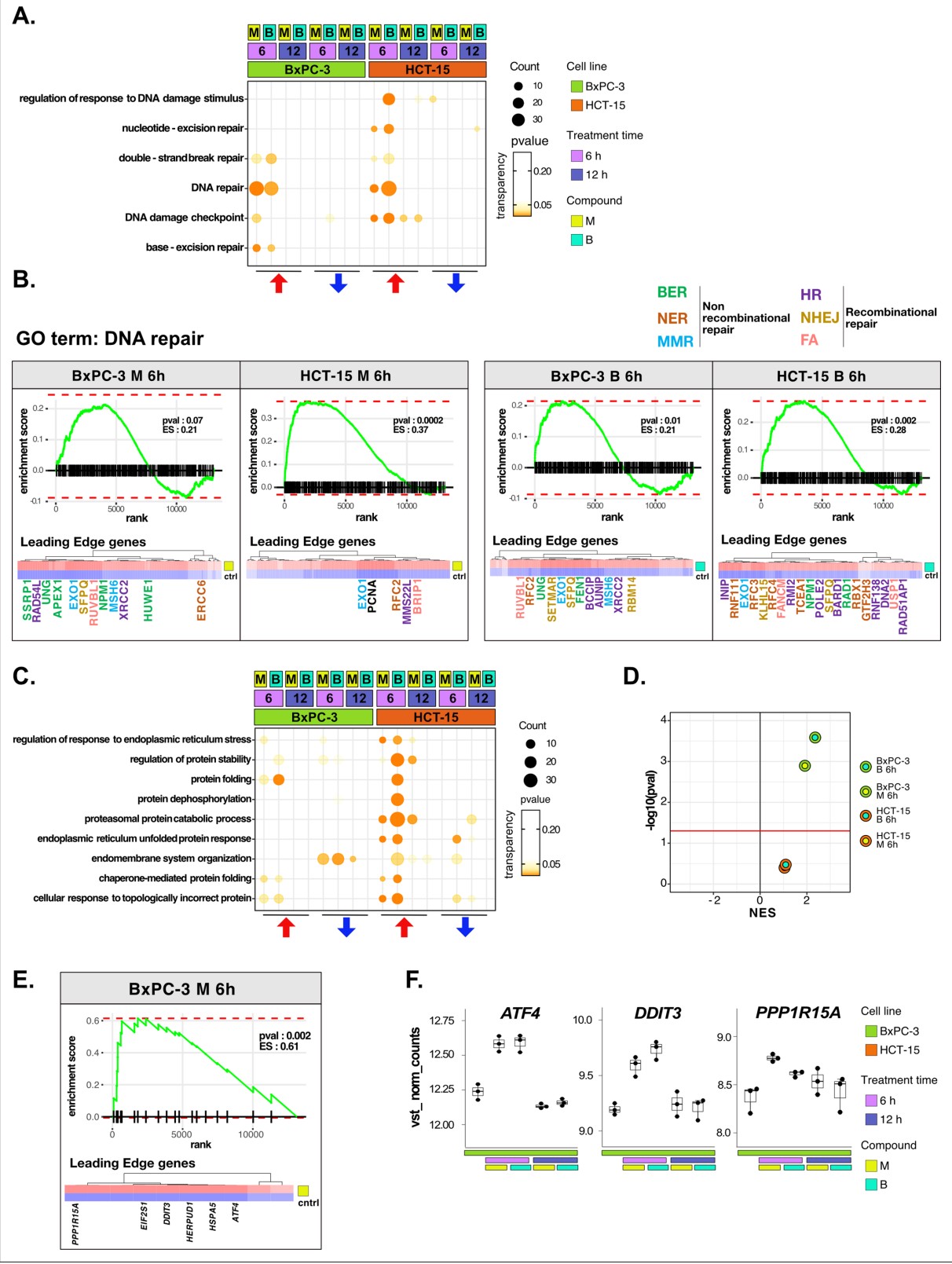

**Figure 3.** DNA repair and proteostasis are key modulators of the response to 3-chloropiperidines (3-CePs). (**A**) Gene set enrichment analysis (GSEA) for terms related to DNA damage and repair performed on differential expression (DE) genes detected in each of the considered treated vs control comparisons. For each gene ontology (GO) term (p<0.05), enrichments in terms of count and p-value are reported. (**B**) GSEA enrichment plots for the DNA repair pathway obtained from log₂FC ranks for each of the considered treated vs control comparisons. The expression of leading edge genes

*Figure 3 continued on next page*

*Figure 3 continued*

is also shown, where key DE genes are reported with the same color of their associated DNA repair pathways (BER = base excision repair, NER = nucleotide-excision repair, MMR = mismatch repair, HR = homologous recombination, NHEJ = non-homologous end joining, and FA = Fanconi anemia pathway) (*Cantor et al., 2001*; *Charles Richard et al., 2016*; *Ferretti et al., 2016*; *Fortini et al., 2003*; *Fousteri and Mullenders, 2008*; *Jaafar et al., 2017*; *Johnson et al., 1999*; *Kunkel and Erie, 2005*; *Liu and Kong, 2021*; *Li and Jin, 2012*; *Li et al., 2009a*; *Lou et al., 2017*; *Lu et al., 2007*; *Mazin et al., 2010*; *McVey et al., 2016*; *Nijman et al., 2005*; *Niraj et al., 2019*; *Overmeer et al., 2010*; *Parsons et al., 2009*; *Pascucci et al., 2018*; *Piwko et al., 2016*; *Poletto et al., 2014*; *Poulsen et al., 2013*; *Prasad et al., 2000*; *Rajendra et al., 2014*; *Satoh and Hanawalt, 1996*; *Smirnova et al., 2005*; *Tellier and Chalmers, 2019*; *Westermark et al., 2003*; *Xu et al., 2008*; *Yard et al., 2016*). (**C**) GSEA for terms related to protein stability and endoplasmic reticulum (ER) load performed on DE genes detected in each comparison. For each GO term, enrichments in terms of count and p-value are reported. (**D**) NES (normalized enrichment score) and $-\log_{10}$ pval for the $\log_2$FC rank-based GSEA enrichment of the GO term PERK-mediated unfolded protein response (UPR) in treated vs control comparisons. (**E**) GSEA enrichment plot for the PERK-mediated UPR pathway obtained from $\log_2$FC rank in the M 6 hr vs control comparison in BxPC-3 cells. The expression of leading edge genes is also shown, where key DE genes of the mentioned pathway are reported. (**F**) Boxplots showing the expression level of *ATF4*, *DDIT3,* and *PPP1R15A* (vst-transformed normalized counts) in BxPC-3 cells (n=3).

The online version of this article includes the following figure supplement(s) for figure 3:

**Figure supplement 1.** DNA repair and proteostasis are key modulators of the response to 3-chloropiperidines (3-CePs)—supporting data.

identified especially in BxPC-3 cells (p<0.05, *Figure 4B*, *Figure 4—figure supplement 1C*). For further downstream analyses, we focused on DARs mapping to promoters, whose specific condensation or compaction contribute to modulation of transcription of associated genes (*Figure 4B*).

Also in this case, to better describe the timing of chromatin remodeling, cell-specific promoter-associated DARs elicited after 6 hr and 12 hr of treatment were grouped in clusters sharing a similar pattern of regulation and functional enrichment was performed on the associated genes (*Figure 4C*, *Figure 4—figure supplement 1D*, *Supplementary file 3*).

In BxPC-3 cells, we observed condensation of promoters involved in protein catabolism and DNA damage detection after 6 hr of exposure (Cluster 1 and 2, *Figure 4C*), most likely contributing to the transcriptional downregulation of such processes observed at the same time point and in line with the reported improper detection of DNA lesions (*Ran et al., 2007*; *Zhou et al., 2017*). On the contrary, relaxation of peaks involved in apoptosis and redox regulation was detected, in line with evidence from RNA-seq (Cluster 5, *Figure 4C*). In HCT-15 cells, relaxation of promoters involved in the DDR as well as protein catabolism (Cluster 9 and 10, *Figure 4C*) was observed, again supporting our observations on transcriptome level. Accordingly, we also evidenced a cluster of downregulated promoters involved in rRNA transcription (Cluster 6, *Figure 4C*). Altogether, these results attested that the regulation of elicited transcriptional pathways was accommodated by changes at the chromatin level, adding new information on the possible mechanisms determining the cellular responses to 3-CePs.

A critical step in the analysis of multi-omic datasets is the integration of information obtained from the different layers. Though valuable strategies have been developed in recent years to integrate RNA- and ATAC-seq data, alternatives are still required to optimize and enlarge the functional information obtained from the combination of these powerful techniques (*Ackermann et al., 2016*; *Höllbacher et al., 2020*; *Yan et al., 2020*). In this study, we approached data integration through two alternative strategies, that we called pairwise and crosswise.

As a first level of integration, we identified genes with concordant regulation in RNA- and ATAC-seq upon treatment. In this pairwise integration, we compared the direction of transcriptional regulation of genes to the accessibility of their promoters, as specified in the Methods section and shown in *Figure 4D* (**M** 6 hr), *Figure 4—figure supplement 2A* (**B** 6 hr), and *Figure 4—figure supplement 2B* (12 hr). Given the biological delay that could exist between chromatin remodeling and a detectable variation in transcript level, pairwise comparisons were also considered between chromatin changes after 6 hr and transcriptional responses after 12 hr of treatment (*Figure 4—figure supplement 2C*). We then applied GSEA to the identified groups of genes across all conditions to point out functional processes of interest.

Overall, the integration efficiently identified key regulators of previously described processes, once more highlighting the value of applying multi-omics to gain a higher resolution and robustness in the analysis of drug responses. As described in *Figure 4D* for compound **M**, and similarly for **B** (*Figure 4—figure supplement 2A*), we observed coherent upregulation of genes involved in ribosome biogenesis (e.g. *RNPS1*), apoptosis (e.g. *AEN*), BER (e.g. *APEX1*), and cell cycle regulation (e.g.

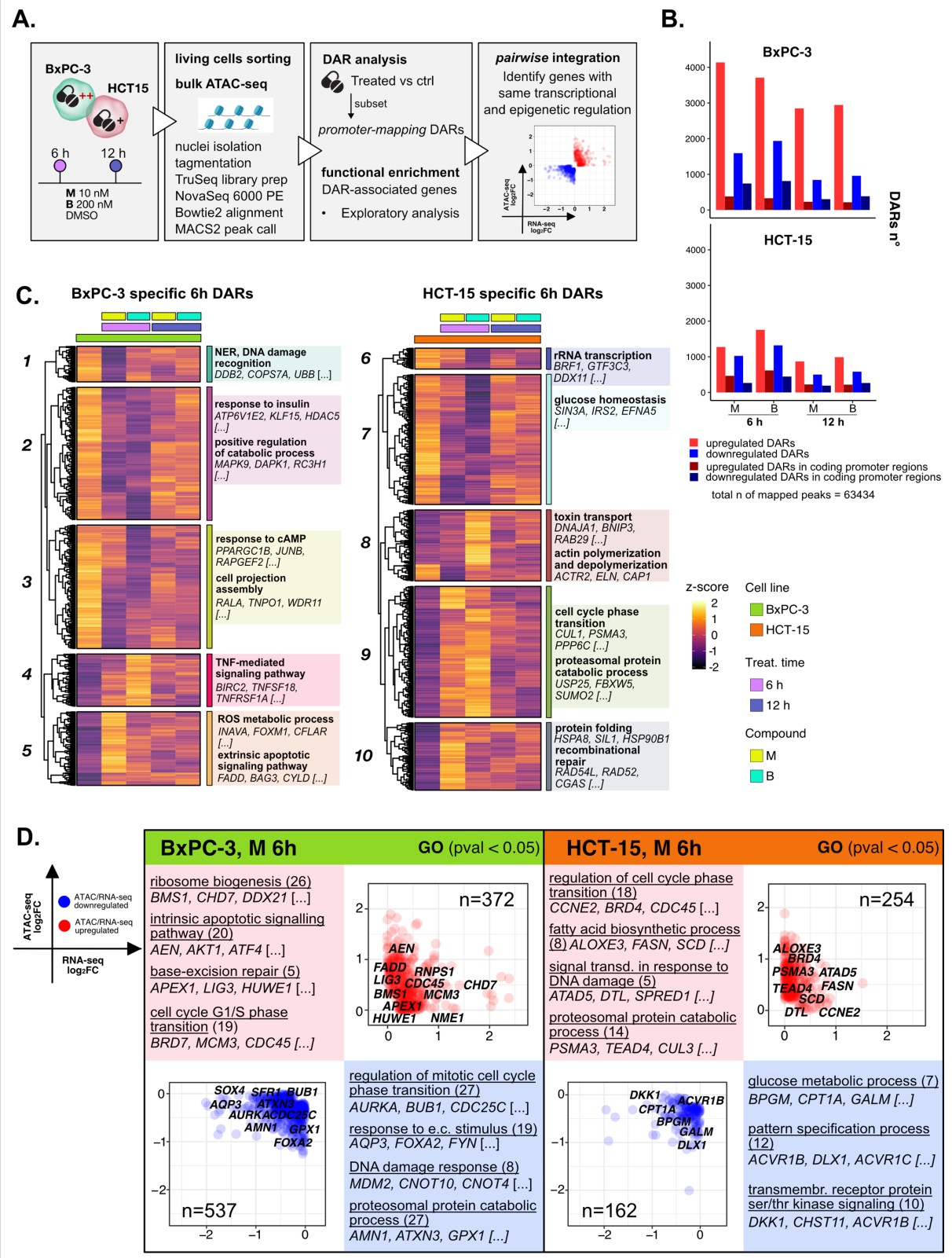

**Figure 4.** The response to 3-chloropiperidines (3-CePs) is further regulated at the chromatin level. (**A**) An overview of the applied workflow for the ATAC-seq analysis. (**B**) Number of up- (red) and down-regulated (blue) differentially accessible regions (DARs) in BxPC-3 and HCT-15 cells after treatment with M (10 nM), B (200 nM), or DMSO 0.5% (ctrl) for 6 hr and 12 hr (p-value threshold = 0.05, shrinkage = TRUE). Light blue/red = all detected DARs, dark blue/red = protein coding DARs mapping in promoter regions. (**C**) Accessibility level of cell-specific 6 hr DARs across test conditions. Gene

*Figure 4 continued on next page*

*Figure 4 continued*

set enrichment analysis (GSEA) was performed on genes associated with DARs with similar regulation, grouped in modules identified by hierarchical clustering: for each cluster, representative gene ontology (GO) terms and genes of the associated load are reported. (**D**) Pairwise integration: ratio-ratio plots report the RNA- and ATAC-seq log$_2$FCs of genes showing the same direction of transcriptional and chromatin accessibility regulation and their GSEA. Integration was performed not only at the same time point of 6 hr in both omic layers. For each GO term (p<0.05), enrichments in terms of count and p-value are reported.

The online version of this article includes the following figure supplement(s) for figure 4:

**Figure supplement 1.** The response to 3-chloropiperidines (3-CePs) is further regulated at the chromatin level—supporting data.

**Figure supplement 2.** The response to 3-chloropiperidines (3-CePs) is further regulated at the chromatin level—supporting data 2.

*CDC45*) at 6 hr of treatment in BxPC-3 cells. Meanwhile, at the same conditions, genes regulating the DDR (e.g. *MDM2*), proteostasis, and redox balance (e.g. *GPX1*) were found to be downregulated consistently in both omic layers.

Instead, the last mentioned processes were activated in HCT-15 cells, as anticipated from previous findings. Indeed, the integration highlighted double-regulated modulators of protein catabolism (e.g. *PSMA3*), DNA repair (e.g. *DTL*), and lipid metabolism, such as the previously described *SCD*. Interestingly, these cells also turned out to activate TGF β signaling (e.g. *BMPR1A*), an important player in cancer drug resistance, in response to **B** (*Figure 4—figure supplement 2A*; *Brunen et al., 2013*). Furthermore, BxPC-3 cells downregulated, early after 6 hr, genes involved in actin remodeling, a mechanism affecting morphology and function of cancer cells (e.g. *CARMIL1*) (*Figure 4—figure supplement 2A*; *Caridi et al., 2019*; *Tafazzoli-Shadpour et al., 2020*). Additional insights on double regulation of further processes were evidenced at 12 hr (*Figure 4—figure supplement 2B*), suggesting for instance autophagy (e.g. *CALCOCO2*) as a putative pathway accounting for the enhanced catabolism in HCT-15 cells (*Chang and Zou, 2020*).

Collectively, pairwise integration of RNA- and ATAC-seq shed light on genes with robust regulation at the transcriptional and chromatin level, adding further details to the previously identified response pathways.

## Crosswise integration expedites the comprehension of multi-omic data

Through the pairwise approach, we identified genes with both transcriptional and chromatin regulation which significantly contributed to the observed cellular response. We further evaluated the crosstalk between RNA- and ATAC-seq at a different level by focusing on groups of genes co-regulated in the two omic layers. The identification of genes sharing similar regulation across conditions either at the transcriptional or chromatin level would maximize the detection of interacting pathways and regulatory processes, e.g., as a result of chromatin changes in promoters tuning the transcription of a certain gene set. This approach, which we termed crosswise integration, was achieved by a combination of horizontal and vertical construction of co-expression network analysis (hCoCena and vCoCena).

Both algorithms are based on CoCena, which inspects the pattern of gene regulation across conditions in a single transcriptome dataset. Its evolution hCoCena can inspect gene regulatory patterns across omic datasets of the same type and integrate them in a single network (*Aschenbrenner et al., 2021*). Instead, vCoCena was designed to define modules of genes and/or genomic markers, such as DARs, with a similar pattern of regulation across the same conditions in different types of omic data, acting as a multi-omics integration approach.

As a first step, we created separate co-expression networks for the RNA- and ATAC-seq layers (*Figure 5A*, *Figure 5—figure supplement 1A*). For each of the omic layers, to prevent the construction of a network mostly describing the difference between the two cell lines, we first created independent networks for BxPC-3 and HCT-15 cells. We then integrated the cell-type specific networks separately for transcriptome and chromatin accessibility using hCoCena (RNA-seq BxPC-3 with RNA-seq HCT-15; ATAC-seq BxPC-3 with ATAC-seq HCT-15) (*Aschenbrenner et al., 2021*). The union of the top 1000 most variable DE and top 1000 variable promoter DAR-associated genes detected in treated conditions was selected as input for constructing individual networks (overall 1919 input genes). Clustering of the resulting RNA- and ATAC-seq networks identified a relevant number of gene modules with highly specific regulatory patterns (*Figure 5—figure supplement 1B and C*). At this point, the vertical, inter-omic integration (vCoCena) was applied to construct the final network consolidating the

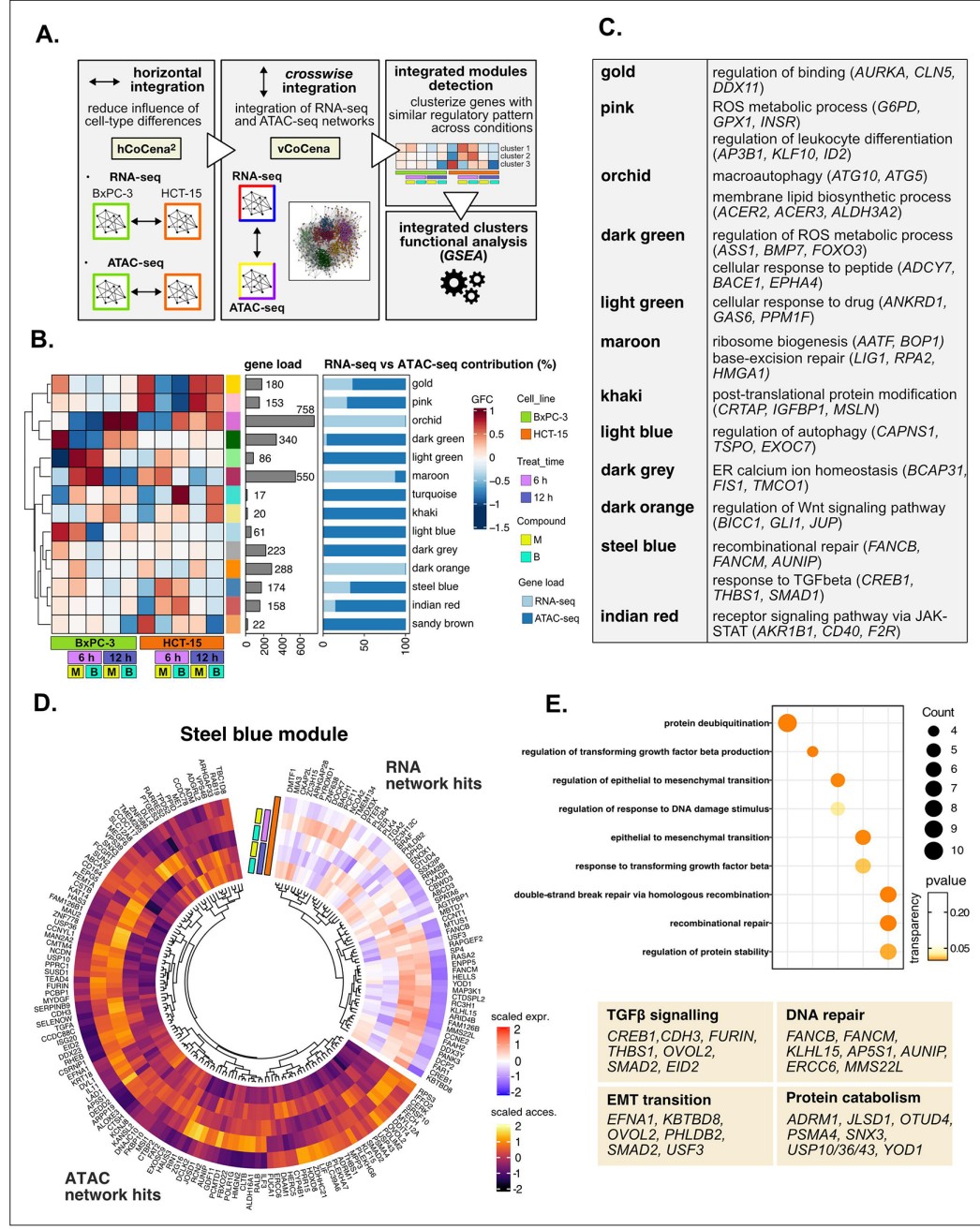

**Figure 5.** Crosswise integration expedites the comprehension of multi-omic data. (**A**) Overview of the applied workflow for the crosswise integration analysis. (**B**) Integrated modules of genes from the RNA- and ATAC-seq layers obtained with vertical construction of co-expression network analysis (vCoCena) and associated group fold change (GFC) pattern of regulation across conditions. The relative contribution of hits from the RNA- or ATAC-seq layers is also reported for each module. (**C**) Representative gene ontology (GO) terms (p<0.05) for the most relevant modules of genes, identified by gene set enrichment analysis (GSEA). Enrichments in terms of count and p-value are reported. (**D**) Expression and chromatin accessibility levels in HCT-15 cells of genes included in the *steelblue* module (nodes can come from the RNA- or ATAC-seq layer). (**E**) Most representative GO terms from GSEA on genes of the *steelblue* module (key areas: EMT (Epithelial-Mesenchymal Transition), protein catabolism, TGF β signaling, and DNA repair). For each GO term (p<0.05), enrichments in terms of count and p-value are reported.

The online version of this article includes the following figure supplement(s) for figure 5:

**Figure supplement 1.** Crosswise integration expedites the comprehension of multi-omic data—supporting data.

information from transcriptome and chromatin accessibility (*Figure 5—figure supplement 1D*, see Methods for details). The new overall network was then reclustered resulting in integrated modules of co-regulation including nodes originally derived from the two separate layers in different ratios, as shown in *Figure 5B*.

The approach combined genes sharing similar regulation in the respective omic dataset, as approximated by the group fold-change (GFC) pattern, with the postulate that genes grouped together cooperate in specific cellular processes. To define the underlying mechanisms, GO enrichment was performed on genes included in each of the modules, and representative biological terms for the most relevant clusters were reported in *Figure 5C*, *Supplementary file 4*. Some modules validated the information obtained through previous analyses (*Figure 5—figure supplement 1E*): both the orchid and light blue clusters suggested macroautophagy as a putative pathway accounting for the enhanced catabolism observed in HCT-15 cells (*Chang and Zou, 2020*). Consistently, the former RNA-seq-based module was downregulated at 6 hr in BxPC-3 but upregulated already after 6 hr with **B** in HCT-15, while the latter ATAC-seq-based module included peaks condensing after 6 hr only in BxPC-3, confirming the latter cell line as refractory to a rapid engagement of its protein catabolism apparatus. Another mostly RNA-seq-based module validating our previous approach was the maroon module, upregulated after 6 hr in BxPC-3, containing genes involved in ribosome biogenesis, BER, and apoptosis. Furthermore, the dark green and dark gray ATAC modules, downregulated in BxPC-3 cells, include genes modulating redox metabolism and ER stress.

Other modules offered instead a new perspective on the reactions of the two cell lines: the dark orange RNA module includes genes downregulated in HCT-15 cells at 6 hr but upregulated in BxPC-3 responding to **M**. Interestingly, these hits belong to the Wnt signaling pathway, well-known for regulating ribosome biogenesis and cell growth, adding further information on our initial findings. In addition, the indian red module, upregulated in HCT-15 cells, points toward the tumorigenic JAK-STAT as an additional pathway modulating drug resistance (*Brooks and Putoczki, 2020*).

The crosswise integration also identified additional regulation, exemplified by the steelblue module. As approximated by the associated GFCs pattern, its 174 genes are positively modulated in HCT-15 cells especially after 6 hr of treatment (*Figure 5D*). Interestingly, functional enrichment identified hits both from RNA- and ATAC-seq involved in the tumorigenic epithelial-mesenchymal transition (*Figure 5E*), a mechanism regulating morphology and invasiveness of cancer cells (e.g. *KBTBD8*, *Figure 5—figure supplement 1F*, from the RNA-seq layer) (*Brabletz et al., 2018*). Other module genes belonged to TGF β signaling (e.g. *FURIN* and *CREB1*; *Figure 5E*), a pathway identified already from the pairwise approach and involved in cancer drug resistance (*Brunen et al., 2013*). In addition, the module included genes of recombinational DNA repair belonging to both omic layers, which is in line with our initial findings (e.g. *AUNIP*; *Figure 5E* and *Figure 5—figure supplement 1F*).

Overall, the crosswise integration of RNA- and ATAC-seq data allowed an efficient combination of the functional information from the two omics layers. Clearly, the whole integration added further biology to what we had identified when analyzing transcriptional and chromatin landscape regulation individually.

## Perturbation-informed basal signatures efficiently predict sensitivity to candidate drugs

The information derived from the crosswise integration was employed to construct a signature of sensitivity to 3-CePs. Being more potent, **M** was selected as reference to describe a sensitivity prediction framework based on the use of a perturbation-informed omic signature (*Figure 6A*, *Figure 6—figure supplement 1A*, Methods).

First, we selected vCoCena clusters based on the highest difference in regulation between the two cell lines after treatment with **M**, considering only the most informative time point of 6 hr (selected modules: maroon, dark green, steelblue, indian red, light green, pink, sandy brown; module selection criteria are described in detail in the Methods section). According to our hypothesis, genes that belong to these modules, coming both from RNA- and ATAC-seq analyses, are expected to be the major determinants of the differential susceptibility in the two cell lines.

Importantly, we postulated that features accounting for sensitivity should be intrinsic for the two cell lines, thus explained already by significant differences in their basal status. For this reason, we performed DE analysis between untreated BxPC-3 and HCT-15 control groups, identifying genes

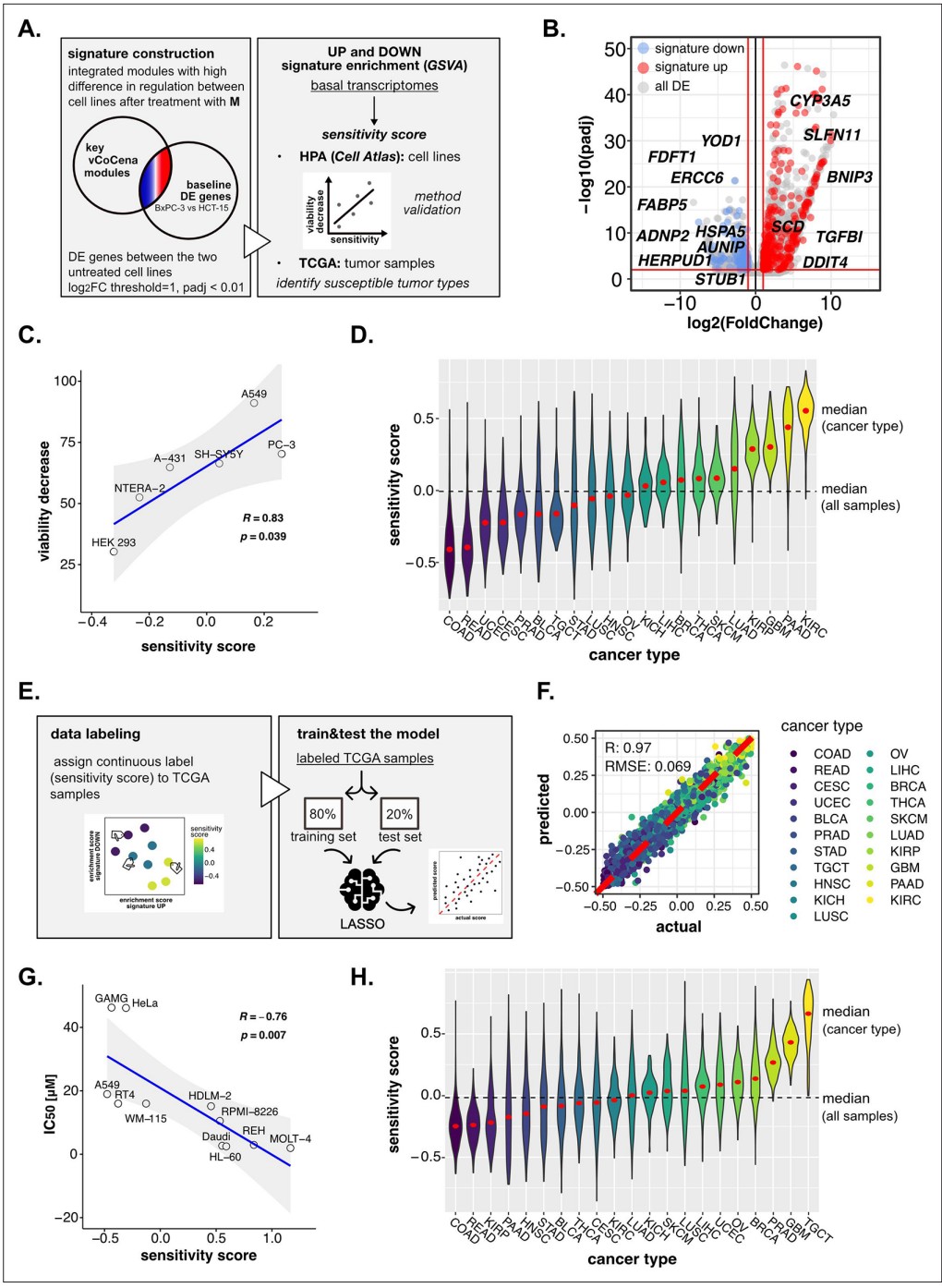

**Figure 6.** Perturbation-informed basal signatures efficiently predict sensitivity to our candidate drugs. (**A**) Overview of the applied workflow for the sensitivity signature construction and associated drug susceptibility prediction. (**B**) M sensitivity signature genes (red = signature up and blue = signature down) pinpointed from all differential expression (DE) genes in the BxPC-3 vs HCT-15 baseline comparison. (**C**) Pearson correlation between predicted sensitivity score and viability decrease in a subset of human protein atlas (HPA) (cell atlas) cell lines (validation set). (**D**) Sensitivity scores predicted from gene set variation analysis (GSVA) enrichment of our up and down signatures in RNA-seq profiles of TCGA tumor samples. Median values for all sample scores and within each tumor type are reported. (**E**) Overview of the applied workflow for the LASSO-based ML setup. (**F**) Predictive outcome of the trained model (Pearson correlation R and RMSE (root-mean-square error) are reported). (**G**) Pearson correlation between predicted cisplatin sensitivity score and IC$_{50}$ in a subset of HPA (cell atlas) cell lines (validation set).

*Figure 6 continued on next page*

*Figure 6 continued*

(**H**) Cisplatin sensitivity scores predicted from GSVA enrichment of our up and down signatures in RNA-seq profiles of TCGA tumor samples. Median values for all sample scores and within each tumor type are reported.

The online version of this article includes the following figure supplement(s) for figure 6:

**Figure supplement 1.** Perturbation-informed basal signatures efficiently predict sensitivity to our candidate drugs—supporting data.

**Figure supplement 2.** Validation analysis on cisplatin sensitivity confirms the versatility of the pipeline.

---

up- and down-regulated at the transcriptional level in the high-sensitive cell line (p<0.01, IHW correction, abs[log$_2$FC]>1), and sorted out only those belonging to previously selected modules. This approach resulted in a subgroup of genes with different basal expression in BxPC-3 cells as well as a sufficiently compound- and cell line-specific regulation upon perturbation. This perturbation-informed signature was composed of 307 genes upregulated (signature up) and 123 genes downregulated (signature down) in the high-sensitive BxPC-3 cells (*Figure 6B*, gene list available in *Supplementary file 5*). GO enrichment on these genes identified protein synthesis, folding and catabolism, lipid metabolism, matrix organization, and actin remodeling among the most significant biological functions (*Figure 6—figure supplement 1B*). Some interesting genes in the up signature were *BNIP3*, proapoptotic, *SCD*, the already discussed regulator of lipid metabolism and SLFN11, involved in DDR and known response biomarker (*Zoppoli et al., 2012*). Among those composing the down signature, we identified *YOD1*, *HERPUD1*, and *HSPA5*, involved in protein homeostasis and ER stress, but also *ERCC6* and *AUNIP* of the DDR (*Figure 6B*; *Ernst et al., 2009*; *Lou et al., 2017*; *Pascucci et al., 2018*; *Schulze et al., 2005*).

To determine the robustness of the obtained signature and its ability to predict sensitivity to **M**, we next performed a gene set variation analysis (GSVA) on publicly available transcriptomes of common cell lines,(*Uhlén et al., 2015*) testing for both the up and down signatures (*Figure 6—figure supplement 1C*). A sensitivity score was calculated for each cell line as the difference between the enrichment scores (ESs) of the up and the down signatures. The predicted rank was validated experimentally on representative cell lines (A-431, A549, HEK-293, NTERA-2, PC-3, and SH-SY5Y), for which we assessed the residual cell viability after 72 hr treatment with **M** 10 nM (see Methods, *Supplementary file 6*). As shown in *Figure 6C*, our perturbation-informed signature demonstrated a strong predictive capacity as attested by the high positive correlation between viability decrease and predicted sensitivity score (Pearson's R: 0.834, p: 0.0398; *Figure 6C*). This signature outperformed a random one containing the same number of genes (R: −0.023, p: 0.97, *Figure 6—figure supplement 1D*) and also a signature of equal size composed by the top up- and down-regulated genes between the two cell lines (R: 0.33, p: 0.52, *Figure 6—figure supplement 1E*, gene list available in *Supplementary file 7*). Collectively, our crosswise integration approach resulted in a perturbation-informed signature capable of predicting drug sensitivity in a wide range of untreated tumor cell lines commonly used in cancer research.

Encouraged by these results, we adapted our strategy to mimic a clinical setting utilizing the primary tumor samples of the Cancer Genome Atlas TCGA database. By applying GSVA, we examined the relative distribution of samples from different tumor types based on the calculated sensitivity score, unveiling which cancer types were predicted as generally more susceptible (i.e. kidney renal clear cell carcinoma [KIRC], PAAD, glioblastoma multiforme [GBM], and kidney renal papillary cell carcinoma [KIRP]) or less sensitive (i.e. rectum adenocarcinoma, colon adenocarcinoma [COAD], uterine corpus endometrial carcinoma, cervical squamous cell carcinoma and endocervical adenocarcinoma) to **M**, providing a framework for further in vivo development of this compound (*Figure 6D*).

Interestingly, the predicted tumor type with the highest sensitivity turned out to be KIRC, demonstrating that the designed signature was not driven by the original cell type of the cell lines used for its extrapolation and could go beyond the original cancer type. At the same time, PAAD and COAD (as BxPC-3 and HCT-15 cells) were still among the most and least sensitive, confirming that cell type intrinsic determinants of susceptibility exist and are represented in our signature. Interestingly, intra-tumor variability resulted in a continuous distribution of samples scores within each cancer group, confirming the importance of clinically translating such predictions beyond the tumor type to better address patient-specific therapeutic needs.

To enlarge the accessibility and clinical translatability of our framework, we finally introduced a LASSO regression model to predict the sensitivity of tumor samples in the external reference dataset (*Figure 6E*). We trained a regression model using TCGA basal transcriptomic profiles labeled with the previously predicted sensitivity scores in order to create a self-supervised system able to emulate the prediction irrespective of the context dataset, detaching the predictive tool from the data space. From a clinical perspective, this further step would allow one to collect a patient basal transcriptome and feed it to the model, not only improving the performance of the prediction but also avoiding any issue related to data sharing since the model itself does not contain any patients' sensitive data.

In detail, TCGA samples were labeled according to the calculated continuous sensitivity scores. Next, the model was trained on 80% of the data and tested on the remaining 20%, which efficiently predicted drug sensitivity within the test samples (R: 0.97, RMSE: 0.069) (*Figure 6F*). Notably, such predictive capacity was maintained even when excluding from the transcriptomes all the signature genes used to define the sensitivity score label of the samples, suggesting the biological robustness of the predictive system (R: 0.97, RMSE: 0.072; *Figure 6—figure supplement 1F*). In fact, while the signature itself was good enough to rank samples based on experimental biological evidence, the model went beyond the initial signature by relying on additional predictive features previously not identified. Moreover, the classifier performed well also on separate cancer types, demonstrating its capacity to address intra-tumor type heterogeneity (*Figure 6—figure supplement 2A*).

As a last validation step to confirm the versatility of our approach, we applied the pipeline to the prediction of cancer cell lines' sensitivity to cisplatin, a thoroughly reported broad-acting chemotherapeutic introduced as a DNA damaging agent (*Kelland, 2007*). We applied the same analysis pipeline as outlined for the 3-CePs. First, we selected Jurkat and BxPC-3 cells as high- and low-sensitive cancer cell lines based on the $IC_{50}$s reported in the Genomics of Drug Sensitivity in Cancer database (Jurkat: 3 μM, BxPC-3: 55 μM) (*Yang et al., 2013*). Then, we acquired RNA- and ATAC-seq profiles after 6 hr of 3 μM cisplatin treatment in both cell lines. Next, we processed data as reported for 3-CePs (*Figure 6—figure supplement 2B*, C, D), up to the construction of an integrated vCoCena network (*Figure 6—figure supplement 2E*). At this point, we built the perturbation-based sensitivity signature specific for cisplatin. Due to the higher number of DE genes between the cell lines using previous parameters, we applied a higher logFC threshold in the call between the untreated controls (p<0.01, IHW correction, abs[logFC]>2.5). In this way, we obtained an overall signature comparable in size to the **M** one (signature up: 107, signature down: 363, see Methods for details, *Supplementary file 8*). The signature was then tested on Human Cell Atlas (HCA) transcriptomes, subsetting for cancer cell lines whose $IC_{50}$ was available in the Genomics of Drug Sensitivity in Cancer database (*Supplementary file 9*). Results confirmed the applicability of our approach, which predicted sensitivity to this reference drug with high accuracy, showing a strong anticorrelation between predicted sensitivity score and actual $IC_{50}$ (Pearsons's R: −0.76, p: 0.007, *Figure 6G*). Even more promising, despite the fact that our model was not trained on data derived from testicular cancer, the prediction on TCGA tumor transcriptomes highlighted testicular germ cell tumor TGCT as the most sensitive cancer type (*Figure 6H*), a finding that is clinically confirmed being testicular cancer efficiently treated with cisplatin since decades (*Kelland, 2007*).

Overall, we demonstrated how to further employ the integrated RNA- and ATAC-seq information to assemble an accurate and clinically-accessible predictive strategy capable of orienting drug development and supporting medical practice in the context of precision oncology.

## Discussion

Despite the advances of the last decades, efforts are required to expedite routine use of omic-scale approaches in preclinical and clinical settings. Recent work illustrated the potential for omics technologies to accelerate the process of drug discovery from the initial identification of candidate lead compounds up to their preclinical and clinical development (*Kagohara et al., 2020*; *Li et al., 2014*; *McFarland et al., 2020*; *Rendeiro et al., 2020*; *Shaheen et al., 2018*; *Srivatsan et al., 2020*; *Ye et al., 2018*). Furthermore, improvements in computational approaches for omics data analyses (*Koromina et al., 2019*; *Li et al., 2021*; *Mun et al., 2020*) and an ever-increasing availability of public reference datasets (*Perez-Riverol et al., 2019*) make it now possible to develop completely new pipelines to address the pharmacological profile of any given drug, from its MoA to sensitivity biomarkers (*Dugger et al., 2018*; *Matthews et al., 2016*; *Paananen and Fortino, 2020*).

Here, we combined transcriptome and chromatin accessibility analyses within perturbation experiments to investigate the specific activity profile of 3-CePs, a new class of potential anticancer agents acting as DNA alkylators (*Carraro et al., 2021*; *Carraro et al., 2019*; *Helbing et al., 2020*; *Sosic et al., 2017*; *Zuravka et al., 2015a*; *Zuravka et al., 2014*; *Zuravka et al., 2015b*). Our analysis unveiled the basis of the preferential activity of 3-CePs against the pancreatic cancer cell line BxPC-3, which was demonstrated to be unable to properly control proteostasis and DDR under stress conditions upon exposure to the alkylating agents. On the contrary, the low-sensitive colorectal adenocarcinoma cell line HCT-15 potentiated protein folding and catabolism all together activating a more efficient DNA repair after treatment. Due to unresolved genotoxic stress and proteostasis dysregulation, widely described as crosstalking events (*González-Quiroz et al., 2020*; *Hutt and Balch, 2010*), BxPC-3 cells activated the apoptotic branch of the PERK-mediated UPR via CHOP and GADD34, both upregulated after treatment (*Han et al., 2013*; *Urra et al., 2013*). Accordingly, such behavior is in line with the described sensitivity of pancreatic cancer adenocarcinoma to ER stress and protein dyshomeostasis (*Garcia-Carbonero et al., 2018*).

Beyond validating the described results, the analysis of chromatin accessibility was first employed to identify genes with concordant transcriptional and epigenetic regulation, a step we called pairwise integration. The approach revealed both epigenetic and transcriptional regulation of key response pathways, identifying apoptotic and ribosome biogenesis mediators upregulated in BxPC-3 and downregulated in HCT-15, as well as redox balance and proteostasis hits upregulated in HCT-15 and downregulated in BxPC-3. In addition, actin dynamics, shown to assist DSBs repair (*Caridi et al., 2019*), were identified to be silenced in the more sensitive cell line.

To further evaluate the interaction between transcriptional and chromatin accessibility responses, we proposed here a new versatile approach for the crosswise integration of RNA- and ATAC-seq, based on vCoCena. This approach identified modules of genes co-regulated in the two omic layers across the analyzed experimental conditions. Combined with the pairwise integration, this standalone method not only recapitulated the result of the independent transcriptomic and epigenomic analysis, but also highlighted additional pathways, e.g., EMT and TGF β signaling, which modulate the response to the compounds. Indeed, a protumorigenic role was established for TGF β in mediating carcinogenicity linked to epithelial-mesenchymal transition, both processes that could additionally explain the more efficient response of HCT-15 cells to 3-CePs (*Brunen et al., 2013*). Efficient and versatile, this approach represents a valid option to integrate the information from multi-omic studies substituting the separate examination of each omic dataset.

To further assist the development of 3-CePs, we set up a pilot sensitivity prediction framework readily transferable from the bench to the clinics. We designed a perturbation-informed signature derived from the integrated omic layers filtering the differentially expressed genes between the two cell lines at a steady state for those specifically involved in the cellular response to the treatment. Though based on a limited number of perturbed profiles, this gene signature predicted with high precision the sensitivity to 3-CePs only relying on the untreated transcriptome of test cell lines. Not only did this approach work when applied to our candidate drugs, but it also efficiently predicted sensitivity to the clinically established antineoplastic agent cisplatin. The possibility to improve predictions from basal transcriptomes sounds attractive from a clinical perspective since it overcomes the need to screen for thousands of drugs and collect the same amount of profiles from limitedly-available patient samples, such as biopsies (*Adam et al., 2020*). Applied to TCGA tumor samples, this approach provided a list of susceptible cancer types, e.g., KIRC and PAAD, to support the further development of our drug candidate, and, once transferred on an ML platform, could offer a versatile predictive strategy translatable to the clinics (*Koromina et al., 2019*; *Warnat-Herresthal et al., 2021*).

In this study, we combined transcriptomic and epigenetic data to guide our exemplary analysis. Nevertheless, the modularity of our framework allows, with only minimal adjustment, its application to other omic technologies or experimental designs. The pipeline showed to be versatile, being applicable to other drugs and, as a consequence, generalizable to other diseases and target cell populations. Indeed, the vCoCena integration, which is instrumental for both the biological interpretation of the data and the definition of the perturbation-informed signature, is agnostic of the type of data used as soon as this is reduced to a network of co-regulation.

In conclusion, we present a complete end-to-end workflow to implement the use of multi-omics in drug development, providing a human-readable toolbox to interrogate pharmacological questions

in both preclinical and clinical settings. We applied this framework to understand the MoA of 3-CePs revealing the cellular determinants of sensitivity to this novel class of drugs and providing additional information for their clinical development as anticancer candidates. Given its versatility, we envision our workflow to be a broadly applicable resource to assist researchers in different steps of the drug discovery and development process.

# Methods

**Key resources table**

| Reagent type (species) or resource | Designation | Source or reference | Identifiers | Additional information |
|---|---|---|---|---|
| Cell line (*Homo-sapiens*) | HCT-15 | ATCC | #CCL-225 | |
| Cell line (*Homo-sapiens*) | BxPC-3 | ATCC | #CRL-1687 | |
| Cell line (*Homo-sapiens*) | A549 | ATCC | #CRM-CCL-185 | |
| Cell line (*Homo-sapiens*) | HEK-293 | ATCC | #CRL-1573 | |
| Cell line (*Homo-sapiens*) | Jurkat | ATCC | #TIB-152 | |
| Cell line (*Homo-sapiens*) | PC-3 | Others (see Cell lines culturing) | | |
| Cell line (*Homo-sapiens*) | NTERA-2 | Others (see Cell lines culturing) | | |
| Cell line (*Homo-sapiens*) | SH-SY5Y | Others (see Cell lines culturing) | | |
| Cell line (*Homo-sapiens*) | A-431 | Others (see Cell lines culturing) | | |
| Antibody | Anti-human H2A.X, (mouse monoclonal, clone 2F3) | Biolegend | #613405 | (FC = 1:25) |
| Peptide, recombinant protein | Tn5 | In-house | | |
| Commercial assay or kit | Foxp3 Transcription Factor Staining Buffer Set | eBioscience | #00-5523-00 | |
| Commercial assay or kit | miRNeasy mini kit | QIAGEN | | |
| Commercial assay or kit | TruSeq stranded total RNA kit | Illumina | | |
| Chemical compound, drug | Pico488 dsDNA quantification reagent | Lumiprobe | | |
| Chemical compound, drug | Propidium Iodide | Sigma | #P4864 | |
| Chemical compound, drug | LIVE/DEAD Near-IR fixable dye | Invitrogen | | |
| Chemical compound, drug | M, B | *Carraro et al., 2019*; *Zuravka et al., 2014* | | |
| Chemical compound, drug | cisplatin | Sigma | | |
| Software, algorithm | FlowJo | BD | | |
| Software, algorithm | bcl2fastq2 | Illumina | | |
| Software, algorithm | STAR | *Dobin et al., 2013* | | |
| Software, algorithm | R | https://www.r-project.org/ | | |
| Software, algorithm | DESeq2 | Bioconductor | | |
| Software, algorithm | clusterProfiler | Bioconductor | | |
| Software, algorithm | Trimmomatic | *Bolger et al., 2014* | | |
| Software, algorithm | bowtie2 | *Langmead and Salzberg, 2012* | | |
| Software, algorithm | Picard | http://broadinstitute.github.io/picard | | |
| Software, algorithm | deeptools | *Ramírez et al., 2014* | | |
| Software, algorithm | samtools | *Ernst et al., 2009* | | |

*Continued on next page*

*Continued*

| Reagent type (species) or resource | Designation | Source or reference | Identifiers | Additional information |
|---|---|---|---|---|
| Software, algorithm | MACS2 | *Lawrence et al., 2013* | | |
| Software, algorithm | GenomicRanges | *Zhang et al., 2008* | | |
| Software, algorithm | GenomicAlignments | *Zhang et al., 2008* | | |
| Software, algorithm | ChIPseeker | Bioconductor | | |
| Software, algorithm | hCoCena | https://github.com/MarieOestreich/hCoCena; *Oestreich, 2022* | | |
| Software, algorithm | vCoCena | This publication | | |
| Software, algorithm | glmnet | *Friedman et al., 2010* | | |

## Cell lines culturing

Colon (HCT-15, #CCL-225), pancreatic (BxPC-3, #CRL-1687), lung (A549, #CRM-CCL-185) carcinoma cell lines, human embryonic kidney (HEK-293, #CRL-1573) and acute T cell leukemia cells (Jurkat, #TIB-152) were purchased from ATCC (American Type Culture Collection). Prostate (PC-3) and testis (NTERA-2) carcinoma cell lines were kindly provided by Prof. W. Kolanus (LIMES institute; University of Bonn), neuroblastoma (SH-SY5Y) by Prof. D. Schmucker (LIMES institute; University of Bonn), and epidermoid (A-431) carcinoma by Prof. G. Zunino (Istituto Nazionale dei Tumori di Milano). Cell lines were maintained in logarithmic phase at 37°C in a 5% carbon dioxide atmosphere using RPMI-1640 (for BxPC-3, HCT-15, PC-3, Jurkat), DMEM (for A-431, HEK-293, NTERA-2, SH-SY5Y), or Ham's F-12K (for A549) media (by Gibco or Euroclone) containing 10% fetal calf serum, antibiotics (50 units/mL penicillin and 50 μg/mL streptomycin) and 2 mM L-glutamine (Euroclone).

## Direct detection and quantification of early DNA damage

The extent of early DNA damage induced by 3-CePs in treated cells was assessed by the Fast Micromethod single-strand-break assay. This approach can detect both single and DSBs, as well as alkali-labile adduct sites in the DNA of treated cells. 5000 cells/well were seeded in 96-well microplates and treated next day for 6 hr with **M** (10 nM and 100 nM), **B** (200 nM and 2 μM), or DMSO 0.5%. After treatment, we measured the effect of double and single-strand breaks on the rate of unwinding of cellular DNA in denaturing alkaline conditions by monitoring the fluorescence of a dye that preferentially binds to dsDNA up to 20 min (Pico488 dsDNA quantification reagent, Lumiprobe). The assay was performed following the protocol of *Schröder et al., 2006*. Two experimental replicates were performed, each one including three technical repeats. Fluorescence signal was acquired by the FLUOstar Omega microplate reader using Omega 5.11 software (BMG LABTECH). The resulting curves based on mean normalized fluorescence values obtained for each treatment and the control (DMSO 0.5%) are reported in *Figure 1B*.

## Cell cycle and flow cytometric H2AX phosphorylation analyses

Possible effects of 3-CePs treatments on the cell cycle distribution of both cell lines were analyzed by FACS, staining cellular DNA with the PI (propidium iodide, Sigma) dye. In addition, we monitored by antibody staining the phosphorylation of histone H2AX, upstream event of the DDR cascade, after 6 hr and 12 hr of treatment in order to investigate the ability of BxPC-3 and HCT-15 cells to detect DSBs. 200,000 cells/well were seeded in 12-well plates and treated next day for 6 hr, 12 hr, or 72 hr with **M** (10 nM), **B** (200 nM), or DMSO 0.5%. Cells were harvested, washed with PBS, fixed, and permeabilized with the Foxp3 transcription factor staining buffer set (eBioscience, cat. #00-5523-00). In detail, cell suspensions were fixed for 1 hr at room temperature with FixBuffer, washed twice with PermBuffer, and stained with anti-human γH2AX AlexaFluor 488 (Biolegend, clone 2F3, cat. #613405) for 1 hr at 4°C. After the first staining, cells were washed first with PermBuffer, then with PBS and stained secondly with PI (30 min, dark). Samples were acquired on a BD Symphony instrument equipped with five lasers (UV, violet, blue, yellow-green, and red), the spectral overlap between the channels were determined with single stained samples using FACSDiva (v 9.1.2). Samples were analyzed in FlowJo

(BD, v 10.7.1). Events were gated first according to FSC-A and SSC-A and cleaned from cell doublets with three consecutive gates (FSC-A vs FSC-H; SSC-A vs SSC-H; and PI-A vs PI-H). The frequency of cells within each phase of the cell cycle was calculated using the PI-A signal with the FlowJo built-in algorithm (Watson model with constrained G2 peak). Three biological replicates were obtained per condition and unpaired two-tailed Student's *t*-test was performed to assess statistical significance (p<0.05).

## Cells treatments for RNA-seq and ATAC-seq experiments

For both RNA- and ATAC-seq analyses, 300,000 cells/well were seeded in 6-well plates and treated next day for 6 hr, 12 hr, or 72 hr with **M** (10 nM), **B** (200 nM), cisplatin (3 µM, Sigma), or DMSO 0.5%. Both for RNA- and ATAC-seq samples, three experimental replicates were obtained for each condition in case of 3-CePs datasets, four replicates were instead produced for the cisplatin datasets.

## RNA-seq experiment

At the end of the treatment, cells were washed, resuspended in 1 mL QIAzol reagent (Qiagen), and stored at −80°C. We extracted the RNA using the miRNeasy mini kit (Qiagen) and checked the RNA integrity and quantity using the tapestation RNA assay on a tapestation 4200 instrument (both from Agilent). We used 750 ng total RNA to generate NGS libraries using the TruSeq stranded total RNA kit (Illumina) following manufacturer's instructions. We checked library size distribution via tapestation using D1000 on a tapestation 4200 instrument (Agilent) and quantified the libraries via Qubit HS dsDNA assay (Invitrogen). We clustered the libraries at 250pM final clustering concentration on a NovaSeq6000 instrument using SP and S2 v1 chemistry (Illumina) and sequenced paired-end 2×50 cycles before demultiplexing using bcl2fastq2 v2.20.

## ATAC-seq experiment

At the end of the treatment, cells were washed, harvested, resuspended in PBS with 1 mM EDTA, stained with the LIVE/DEAD Near-IR fixable dye (Invitrogen, cat. #10119) for 10 min at 4°C, centrifuged and suspended in PBS with 1 mM EDTA. 20,000 living cells/sample were sorted by FACS and further processed for nuclei isolation and transposition reaction following the protocol of *Buenrostro et al., 2013*. Being the number of dead cells found out to be negligible, sorting before transposition was not applied to samples treated with cisplatin.

We generated ATAC-libraries from tagmented cells following the protocol of Buenrostro et al. We checked library size distribution via tapestation using D5000 assay (ATAC) on a Tapestation 4200 instrument (both from Agilent) and quantified the libraries via Qubit HS dsDNA assay (Invitrogen). We clustered the libraries at 250pM final clustering concentration on a NovaSeq6000 instrument using SP and S2 v1 chemistry (Illumina) and sequenced paired-end 2×50 cycles before demultiplexing using bcl2fastq2 v2.20.

## RNA-seq data analysis

Reads were aligned and quantified with STAR (v 2.5.2 a) (*Dobin et al., 2013*) using standard parameters and mapped against the GRCh38p13 human reference genome (Genome Reference Consortium). Raw counts were imported, prefiltered to exclude low-count genes (3-CePs:>100 reads, 17.693 transcripts; cisplatin:>10 reads, 27.404 transcripts), normalized and VST-transformed (variance stabilizing transformation) following the DESeq2 (Bioconductor, v 1.26.0) pipeline using default parameters (*Gentleman et al., 2004*; *Love et al., 2014*). SVA (surrogate variable analysis) was applied to identify latent variables responsible for batch effects and included in the DESeq2 model (3-CePs: 4 SVs, cisplatin: 3 SVs) (*Leek et al., 2012*). All present transcripts were used as input for PCA. The call for differentially expressed genes was performed for all treated vs control comparisons (separate cell lines) using an adjusted p-value threshold equal to 0.05, where IHW was adopted for multiple testing and shrinkage was applied. Only protein-coding hits were considered for further functional analyses on DE genes. GSEA based on the GO (gene ontology) biological process database was employed for functional enrichments, both based on DE genes (*Supplementary files 1 and 2*) or log$_2$FC-based ranks. All enrichment dotplots report the count and p-value associated with each term, when p<0.05. Representative enrichment terms in *Figure 2C* were selected manually from enrichment maps obtained for each group of genes depicted in the dotplot (*Supplementary file 10*): to remove

semantic redundancy, only the most significant nodes among those converging into the same hub were reported (higher count and lower p-value, example in *Figure 2—figure supplement 1F*). SVA batch-corrected normalized VST-transformed counts were used as input for boxplots, heatmaps, and log$_2$FC-based GSEA. Hierarchical clustering was applied to identify blocks of DE genes with similar regulations across conditions as reported in the presented heatmaps (*Figure 2D*, *Figure 2—figure supplement 2A*). In the same heatmaps, row-scaled expression levels of cell-specific DE genes elicited at 6 hr and 12 hr were reported separately for each of the analyzed conditions.

## ATAC-seq data analysis

After adapter trimming using Trimmomatic v 0.36 (*Bolger et al., 2014*), the sequencing reads were aligned bowtie2 v 2.3.5 against the GRCh38p13 human reference genome (*Langmead and Salzberg, 2012*). Subsequently, duplicated reads were removed using Picard *dedup* function, and the transposase-induced offset was corrected using the deeptools v 3.1.3 *alignmentSieve* function (*Ramírez et al., 2014*). After sorting and indexing bam files with samtools v 1.9 (*Li et al., 2009b*), peak calling was performed using MACS2 v 2.1.2 (*Lawrence et al., 2013*). Peak regions from sample-specific peak calling results were unified in R v 3.6.2 using the *reduce* function implemented in the GenomicRanges package v 1.38.0 (*Zhang et al., 2008*) prior to quantification of sequencing reads in these unified peak regions using the *summarizeOverlaps* function implemented in the GenomicAlignments package v 1.22.1 (*Zhang et al., 2008*). Raw counts were prefiltered to exclude low-count peaks (<20 reads, 3-CePs: 63.434 peaks, cisplatin: 170.844 peaks), normalized and VST-transformed following the DESeq2 (Bioconductor, v 1.26.0) pipeline using default parameters (*Gentleman et al., 2004*; *Love et al., 2014*). Peak regions were annotated using *ChIPseeker* (Bioconductor, v1.22.1). All present peaks were used as input for PCA. The call for DARs was performed for all treated vs control comparisons (separate cell lines) considering a p<0.05 threshold. Only peaks mapping in promoters of protein-coding regions (according to ChIPseeker annotation) were considered for further functional analyses. GSEA based on the GO biological process database was employed for functional enrichments based on DAR-associated genes (*Supplementary file 3*). Normalized and, where specified, vst-transformed counts were used as input for heatmaps and boxplots. Hierarchical clustering was applied to identify blocks of DAR-associated genes with similar regulations across conditions as reported in the presented heatmaps (*Figure 4C*, *Figure 4—figure supplement 1D*). In the same heatmaps, row-scaled accessibility levels of cell-specific DARs at 6 hr and 12 hr were reported separately for each of the analyzed conditions. Reference promoter peaks in case of multiple mapping to the same gene were chosen based on higher variance. For the pairwise integration between transcriptional and chromatin accessibility data, we identified hits having the same sign of regulation in RNA- and ATAC-seq which were DE (protein-coding) and/or DAR-associated (protein-coding mapping in promoters). Reference peaks in case of multiple mapping to the same gene were chosen based on lower p-value in the specific comparison. Since a delay could exist between a prior chromatin remodeling and a detectable variation in the respective transcript level, pairwise comparisons were considered not only at the same time point in both omic layers but also between chromatin changes at 6 hr and transcriptional responses at 12 hr. Interesting gene names for each of the considered comparisons were also reported, and GSEA enrichment was applied as specified above.

## Crosswise integration of RNA-seq and ATAC-seq data

The crosswise integration of transcriptomic and chromatin accessibility data was achieved through an adaptation of the CoCena (construction of co-expression network analysis - automated) tool which inspects the pattern of regulation of genes across conditions in a single transcriptome dataset. The core principles driving both network construction and gene modules detection by CoCena have been described previously (*Aschenbrenner et al., 2021*). In this analysis, we first optimized the design of separate co-expression networks for the RNA- and ATAC-seq layers. To avoid the creation of networks mostly describing cell type differences, we calculated separate networks for BxPC-3 and HCT-15 cells which were then integrated horizontally through hCoCena (*Aschenbrenner et al., 2021*). The union of all top 1000 variable DE and top 1000 variable promoter DAR-associated genes detected in treated conditions was selected as input for constructing all networks. For the construction of cell-specific networks, the specified Pearson correlation cutoffs, edges and nodes for RNA-seq (3-CePs: BxPC-3 cutoff = 0.75, edges = 104002, and nodes = 1711; HCT-15 cutoff = 0.707, edges = 54222, and nodes

= 1747; cisplatin: BxPC-3 cutoff = 0.863, edges = 107434, and nodes = 1766; Jurkat: cutoff = 0.955, edges = 21708, and nodes = 1735) and ATAC-seq (3-CePs: BxPC-3 cutoff = 0.693, edges = 7092, and nodes = 1559; HCT-15 cutoff = 0.707, edges = 5431, and nodes = 1510; cisplatin: BxPC-3 cutoff = 0.783, edges = 297385, and nodes = 1685; Jurkat cutoff = 0.876, edges = 6140, and nodes = 1620) were used. The horizontally integrated networks contained the union of all nodes and edges coming from parent networks, where edges between nodes connected in both parent layers were recalculated as a mean of their original weights. Clustering of the resulting RNA- and ATAC-seq networks was performed based on the infomap (RNA-seq network 3-CePs and ATAC-seq network cisplatin) or walktrap (ATAC-seq network 3-CePs and RNA-seq cisplatin) algorithm, where a threshold of minimum of 15 nodes per cluster was applied (*Figure 5—figure supplement 1B, C*; *Rosvall et al., 2009*). To select the threshold, we inspected different values from a minimum of 10 and chose such threshold empirically with the overall goal to maximize the number of detected modules with specific regulation patterns (also in horizontally integrated networks) without compromising the possibility to get efficient functional enrichments.

Subsequently, inter-omic integration by vCoCena was applied to construct the final network. In this case, the correlation between the mean GFC pattern of modules belonging to the two layers was calculated to identify clusters of genes with similar regulation, suitable for crosswise integration. The GFCs are obtained by calculating the mean expression of a gene over all conditions and then computing the fold change of the mean gene expression within each condition from the overall mean (*Aschenbrenner et al., 2021*). Edges from the two separate networks were selected for contributing to the integrated one based on a minimum cross-layer correlation which could guarantee the maximum mixture between layers in identified module pairs (3-CePs: minimum correlation cutoff = 0.665, edges = 206200, and nodes = 3030; cisplatin: minimum correlation cutoff = 0.69, edges = 796517, and nodes = 3104). The new network was reclustered exploiting the walktrap (3-CePs) or infomap (cisplatin) algorithm, applying the same threshold of a minimum of 15 nodes per cluster, and mean GFCs were recalculated: the resulting integrated modules included nodes originally derived from the two separate layers in different ratios, as shown in the relative heatmap (*Figure 5B*, *Figure 6—figure supplement 2E*). Clustering algorithms were chosen empirically based on the evaluation of the resulting modules in terms of size, represented regulatory complexity/diversification as well as functional validity. GO-based GSEA was performed on detected modules of genes (*Supplementary file 4*) and the most significant terms (p<0.05) were reported.

## Sensitivity signature construction and prediction pipeline

For the signature of sensitivity to **M** and cisplatin, relevant modules from the crosswise vCoCena integration were selected as follows (*Figure 6—figure supplement 1A*): for each module, in both cell lines separately, we calculated the difference between the GFC of the control and the **M** (or cisplatin) 6 hr treated groups (ΔGFC [cell line]=GFC(**M**6h) − GFC [ctrl]). The early time point was selected to guide the signature construction since from upstream analyses it turned out to be the most informative of cell responses to 3-CePs. The threshold score was then calculated as the difference between the previously obtained ΔGFCs for the two cell lines (thr$_{score}$ = ΔGFC [BxPC-3] − ΔGFC [HCT-15]). Modules with thr$_{score}$ above q50, thus modules where the regulation was sufficiently different in the two cell lines after treatment with **M**, were selected (**M**: maroon, dark green, steelblue, indian red, light green, pink, sandy brown; cisplatin: orchid, maroon, dark orange, gold, indian red, light blue, and khaki). Genes from the identified modules were grouped together and further considered to drive the definition of our signature of interest.

Further on, DE analysis was performed between BxPC-3 and HCT-15 untreated control groups to identify baseline DE genes up- and down-regulated in the high-sensitive cell line (abs[log$_2$FC]>1 [**M**] or >2.5 [cisplatin], padj <0.01). In fact, given the much higher availability and clinical spendability of RNA-seq compared to ATAC-seq profiles, the signature was finally constructed only from basal transcriptomes. In particular, we further selected among the identified module genes only those that were also DE between the two untreated controls, ending up with a restricted group of genes showing compound- and cell line-specific regulation upon perturbation but, meanwhile, a significantly different basal expression in BxPC-3 cells. This perturbation-informed signature for **M** was composed of 307 genes upregulated (signature up) and 123 genes downregulated (signature down) in the high-sensitive BxPC-3 cells (listed in *Supplementary file 5*). For cisplatin, the signature was composed by

107 genes upregulated (signature up) and 363 genes downregulated (signature down) in the high-sensitive Jurkat cells (*Supplementary file 8*).

To validate the predictive performance of the obtained signature, GSVA was performed both with up and down signatures on the basal RNA-seq profiles of cancer cell lines included in the HPA (human protein atlas) (*Uhlén et al., 2015*). A sensitivity score was calculated for each cell line as the difference between the ESs of the up and the down signatures. The predicted rank was validated on selected cell lines (A-431, A549, HEK-293, NTERA-2, PC-3, and SH-SY5Y) as described in the next paragraph, and Pearson correlation between predicted sensitivity scores and viability decrease in cells treated with **M** 10 nM for 72 hr was calculated. Two control signatures of the same size were also tested: (1) a random genes signature (composed by random genes among those annotated in the RNA-seq profile of HPA cell lines) and (2) a control signature composed by the top up- and down- $\log_2$FC DE genes between the two cell lines (listed in *Supplementary file 7*). The cisplatin signature was also tested on HCA transcriptomes, subsetting for cancer cell lines whose $IC_{50}$ was available in the Genomics of Drug Sensitivity in Cancer database. Only cell lines with $IC_{50}$ no higher than the one of low-sensitive BxPC-3 cells (55 µM) were included in the correlation plot.

GSVA (v 1.38.2) was applied also on basal transcriptomes of samples from the Cancer Genome Atlas TCGA database, and their sensitivity score was calculated as previously indicated. The relative distribution of samples from different tumor types in terms of calculated sensitivity score was plotted together with the indicated median value for each group, to identify possibly more susceptible tumor types.

Finally, the signature-based prediction was used to train a LASSO-based classifier (*cv.glmnet* function in *glmnet* package v 4.1 to assess lambda penalty, predict function in stats package v 4.0.3 for actual prediction) (*Friedman et al., 2010*). Briefly, TCGA samples were assigned to a continuous label based on the previously inferred sensitivity scores. We next trained the classifier with 80% of these profiles and tested it on the remaining 20%: Pearson correlation and RMSE were calculated to evaluate the predictive performance of the classifier. The classifier was also tested on separate tumor types to assess accuracy in predicting intra-tumor heterogeneity. To assess the biological robustness of our signature and of the obtained model, the classifier was trained and tested also using transcriptomes cleaned up from genes belonging to our signature.

## Validation of 3-CePs sensitivity prediction on cancer cell lines

The rank of sensitivity to **M** obtained from the newly constructed signature was validated on a subset of available cell lines included in the Human Cell Atlas. The selected cell lines spanned quite well between the max and min detected susceptibility scores. Here are the screened cell lines from the one predicted as most sensitive: PC-3, A549, A-431, SH-SY5Y, NTERA-2, and HEK-293. 5000 cells/well were seeded in 96-well microplates and after 24 hr treated with **M** 10 nM for 72 hr. Cell viability was assessed at the end of the treatment by MTT, following previously adopted protocols (*Carraro et al., 2019*). Mean values of residual viability and SDs obtained from two independent experiments in duplicated microplates, each one containing three technical replicates, are reported in *Supplementary file 6*. Pearson correlation between mean residual viability and predicted susceptibility score in considered cell lines was calculated and reported in *Figure 6*.

## Statistics and reproducibility

Unpaired two-tailed Student's *t*-test was performed to assess statistically significant differences ($p < 0.05$) in cell cycle and H2AX phosphorylation analyses between treated and control conditions (n=3). All correlation coefficients were calculated with a Pearson's test. The adopted statistical tests, the considered significance levels, and the number of biological replicates are also reported in figure legends. Boxplots are in the style of Tukey, where the center of the box represents the median of values, hinges represent the 25th and 75th percentile, and the whiskers are extended not further than the 1.5×IQR (inter quartile range). The analysis was performed on R (v. 3.6.2 or 4.0.3): the specific packages used for the analysis, their version and relevant parameters used are explained in the Methods sections. All plots were generated with *ggplot* (v. 3.3.2) except for the heatmaps which were generated with the R package *complexheatmap* (v. 2.2.0). To ensure the reproducibility of the manuscript results, all the analyses were conducted within a containerized environment (Docker). RNA- and ATAC-seq analyses were performed with the docker image jsschrepping/r_docker:jss_R362

(https://hub.docker.com/r/jsschrepping/r_docker). The rest of the analysis was conducted with the image lorenzobonaguro/cocena:v3 (https://hub.docker.com/r/lorenzobonaguro/cocena) for compatibility with the CoCena pipeline.

## Data availability

All raw data included in this study are available at gene expression omnibus (GEO). For 3-CePs, raw RNA-seq data and count matrix are under the GEO accession number GSE179057. Raw ATAC-seq data and peak matrix are available under the accession number GSE179059. Both datasets are collected in a GEO SuperSeries (GSE179064). Cisplatin RNA-seq data and count matrix are available at the accession number GSE207611, the raw ATAC-seq data and peak matrix are available under the number GSE207607. Both are collected in a GEO SuperSeries (GSE207612).The cell line expression data employed in the prediction pipeline were downloaded from https://www.proteinatlas.org/about/download. The file RNA HPA cell line gene data contains transcript expression levels summarized per gene in 69 cell lines and is based on the Human Protein Atlas version 20.0 and Ensembl version 92.38.Similarly, the TCGA expression data from cancer cell samples (the Cancer Genome Atlas) were downloaded from the same web page of the Human Cell Atlas (Transcript expression levels summarized per gene in 7932 samples from 17 different cancer types). Data are based on The Human Protein Atlas version 20.0 and Ensembl version 92.38. Cisplatin $IC_{50}$s in different cell lines can be accessed through the Genomics of Drug Sensitivity in Cancer database (https://www.cancerrxgene.org/compound/Cisplatin/1005/overview/ic50) (*Yang et al., 2013*). Codes to reproduce both pre-processing and downstream analyses reported in this manuscript are available at the public repository https://github.com/ccarraro/3-CePs_prediction, (*Carraro, 2022* copy archived at swh:1:rev:16c71c7857a4980f6ce2a054994d62b78a991e0d). The hCoCena script is accessible at https://github.com/MarieOestreich/hCoCena (*Oestreich, 2022*, copy archived at swh:1:rev:0ff2b-77fa7371ce88801b5798dd3c87e0b03d2b1) (vertical integration code available upon request). Supplementary Data are available as .xlsx or .pdf files.

## Acknowledgements

Funding HGF Helmholtz AI grant Pro-Gene-Gen (ZT-I-PF5-23) Marie Oestreich German Research Foundation (DFG) Excellence Strategy (EXC2151-390873048) German Research Foundation (DFG) - SCHU 950/8–1 German Research Foundation (DFG) - GRK 2168 German Research Foundation (DFG) - TP11 German Research Foundation (DFG) - SFB704 BMBF-funded excellence project Diet–Body–Brain (DietBB) EU project SYSCID under grant number 733100 Joachim L Schultze. Joachim L Schultze and Anna C Aschenbrenner are also funded in part by the DFG-funded CRC SFB 1454 Metaflammation, Project number 432325352.

## Additional information

### Funding

| Funder | Grant reference number | Author |
| --- | --- | --- |
| Helmholtz Association | HGF Helmholtz AI grant Pro-Gene-Gen (ZT-I-PF5-23) | Marie Oestreich |
| Deutsche Forschungsgemeinschaft | German Research Foundation (DFG) Excellence Strategy (EXC2151-390873048) | Joachim L Schultze |
| Deutsche Forschungsgemeinschaft | German Research Foundation (DFG) - SCHU 950/8-1 | Joachim L Schultze |
| Deutsche Forschungsgemeinschaft | German Research Foundation (DFG) - GRK 2168 | Joachim L Schultze |

| Funder | Grant reference number | Author |
|---|---|---|
| Deutsche Forschungsgemeinschaft | German Research Foundation (DFG) - TP11 | Joachim L Schultze |
| Deutsche Forschungsgemeinschaft | German Research Foundation (DFG) - SFB704 | Joachim L Schultze |
| Bundesministerium für Bildung und Forschung | BMBF-funded excellence project Diet-Body-Brain (DietBB) | Joachim L Schultze |
| Horizon 2020 Framework Programme | EU project SYSCID under grant number 733100 | Joachim L Schultze |
| Deutsche Forschungsgemeinschaft | German Research Foundation (DFG) CRC SFB 1454 Metaflammation - Project number 432325352 | Joachim L Schultze Anna C Aschenbrenner |

The funders had no role in study design, data collection and interpretation, or the decision to submit the work for publication.

#### Author contributions

Caterina Carraro, Conceptualization, Data curation, Formal analysis, Validation, Investigation, Visualization, Methodology, Writing – original draft, Project administration, Writing – review and editing; Lorenzo Bonaguro, Conceptualization, Data curation, Supervision, Investigation, Methodology, Writing – review and editing; Jonas Schulte-Schrepping, Stefanie Warnat-Herresthal, Formal analysis, Methodology, Writing – review and editing; Arik Horne, Michele De Franco, Investigation, Methodology, Writing – review and editing; Marie Oestreich, Software, Methodology, Writing – review and editing; Tim Helbing, Resources, Writing – review and editing; Kristian Haendler, Methodology, Writing – review and editing; Sach Mukherjee, Anna C Aschenbrenner, Conceptualization, Writing – review and editing; Thomas Ulas, Conceptualization, Software, Methodology, Writing – review and editing; Valentina Gandin, Richard Goettlich, Conceptualization, Resources, Writing – review and editing; Joachim L Schultze, Barbara Gatto, Conceptualization, Resources, Supervision, Funding acquisition, Project administration, Writing – review and editing

#### Author ORCIDs

Caterina Carraro http://orcid.org/0000-0002-3039-4675
Lorenzo Bonaguro http://orcid.org/0000-0001-9675-7208
Tim Helbing http://orcid.org/0000-0002-1284-3254
Anna C Aschenbrenner http://orcid.org/0000-0002-9429-5457
Barbara Gatto http://orcid.org/0000-0001-9465-6913

#### Decision letter and Author response

Decision letter https://doi.org/10.7554/eLife.78012.sa1
Author response https://doi.org/10.7554/eLife.78012.sa2

## Additional files

#### Supplementary files

• Supplementary file 1. Gene ontology (GO) enrichment results for differential expression (DE) genes sets reported in *Figure 2C* (BxPC-3 specific upregulated, HCT-15 specific upregulated, shared upregulated, BxPC-3 specific downregulated, HCT-15 specific downregulated, shared downregulated).

• Supplementary file 2. Gene ontology (GO) enrichment results for genes belonging to the specified heatmap clusters reported in *Figure 2D* (BxPC-3 and HCT-15 cells DE genes after 6 hr of treatment) and *Figure 2—figure supplement 2A* (BxPC-3 and HCT-15 cells differential expression DE genes after 12 hr of treatment).

• Supplementary file 3. Gene ontology (GO) enrichment results for differentially accessible region (DAR)-associated genes belonging to the specified heatmap clusters reported in *Figure 4C* (BxPC-3 and HCT-15 cells DAR genes after 6 hr of treatment) and *Figure 4—figure supplement 1D* (BxPC-3

and HCT-15 cells DAR-associated genes after 12 hr of treatment).

• Supplementary file 4. Gene ontology (GO) enrichment results for detected vertical construction of co-expression network analysis (vCoCena) modules reported in *Figure 5B*.

• Supplementary file 5. Up and down sensitivity signature gene list for compound M.

• Supplementary file 6. Average viability decrease in cell lines treated with M 10 nM for 72 hr with associated SD. For each cell line, predicted sensitivity scores based on our perturbation-informed signature (signature SS), a random one (random SS), a control one based on top up and down $log_2FC$ differential expression (DE) genes between BxPC-3 and HCT-15 (topFC SS) were also reported.

• Supplementary file 7. Top up and down $log_2FC$ control sensitivity signature gene list used as control.

• Supplementary file 8. Up and down sensitivity signature gene list for cisplatin.

• Supplementary file 9. Reported $IC_{50}$ for cell lines treated with cisplatin (Genomics of Drug Sensitivity in Cancer database). For each cell line, predicted sensitivity scores based on our perturbation-informed signature are reported (signature SS).

• Supplementary file 10. Enrichment maps obtained for each group of genes depicted in the *Figure 2C* dotplot.

• Transparent reporting form

### Data availability

All raw data included in this study are available at gene expression omnibus (GEO). For 3-CePs, raw RNA-seq data and count matrix are under the GEO accession number GSE179057. Raw ATAC-seq data and peak matrix are available under the accession number GSE179059. Both datasets are collected in a GEO SuperSeries (GSE179064). Cisplatin RNA-seq data and count matrix are available at the accession number GSE207611, the raw ATAC-seq data and peak matrix are available under the number GSE207607. Both are collected in a GEO SuperSeries (GSE207612). The cell line expression data employed in the prediction pipeline were downloaded from https://www.proteinatlas.org/about/download. The file RNA HPA cell line gene data contains transcript expression levels summarized per gene in 69 cell lines and is based on the Human Protein Atlas version 20.0 and Ensembl version 92.38. Similarly, the TCGA expression data from cancer cell samples (the Cancer Genome Atlas) were downloaded from the same web page of the Human Cell Atlas (Transcript expression levels summarized per gene in 7932 samples from 17 different cancer types). Data are based on The Human Protein Atlas version 20.0 and Ensembl version 92.38. Cisplatin $IC_{50}$s in different cell lines can be accessed through the Genomics of Drug Sensitivity in Cancer database (https://www.cancerrxgene.org/compound/Cisplatin/1005/overview/ic50) (Yang et al., 2013). Codes to reproduce both pre-processing and downstream analyses reported in this manuscript are available at the public repository https://github.com/ccarraro/3-CePs_prediction (copy archived at swh:1:rev:16c71c7857a4980f6ce2a054994d-62b78a991e0d). The hCoCena script is accessible at https://github.com/MarieOestreich/hCoCena (copy archived at swh:1:rev:0ff2b77fa7371ce88801b5798dd3c87e0b03d2b1) (vertical integration code available upon request). Supplementary Data are available as .xlsx or .pdf files.

The following datasets were generated:

| Author(s) | Year | Dataset title | Dataset URL | Database and Identifier |
|---|---|---|---|---|
| Carraro C, Bonaguro L, Schultze JL, Gatto B | 2021 | Perturbation-informed signatures from crosswise integration of transcriptome and chromatin accessibility analyses predict susceptibility to candidate anticancer drugs | https://www.ncbi.nlm.nih.gov/geo/query/acc.cgi?acc=GSE179064 | NCBI Gene Expression Omnibus, GSE179064 |

*Continued on next page*

*Continued*

| Author(s) | Year | Dataset title | Dataset URL | Database and Identifier |
|---|---|---|---|---|
| Carraro C, Bonaguro L, Schultze JL, Gatto B | 2022 | Perturbation-informed signatures from crosswise integration of transcriptome and chromatin accessibility analyses predict susceptibility to candidate anticancer drugs (cisplatin validation dataset) | https://www.ncbi.nlm.nih.gov/geo/query/acc.cgi?acc=GSE207612 | NCBI Gene Expression Omnibus, GSE207612 |
| Carraro C, Bonaguro L, Schultze JL, Gatto B | 2022 | Perturbation-informed signatures from crosswise integration of transcriptome and chromatin accessibility analyses predict susceptibility to candidate anticancer drugs [RNA-seq] | https://www.ncbi.nlm.nih.gov/geo/query/acc.cgi?acc=GSE179057 | NCBI Gene Expression Omnibus, GSE179057 |
| Bonaguro L, Schultze JL, Gatto B | 2022 | Perturbation-informed signatures from crosswise integration of transcriptome and chromatin accessibility analyses predict susceptibility to candidate anticancer drugs [ATAC-seq] | https://www.ncbi.nlm.nih.gov/geo/query/acc.cgi?acc=GSE179059 | NCBI Gene Expression Omnibus, GSE179059 |
| Carraro C, Bonaguro L, Schultze JL, Gatto B | 2022 | Perturbation-informed signatures from crosswise integration of transcriptome and chromatin accessibility analyses predict susceptibility to candidate anticancer drugs (cisplatin validation dataset) [ATAC-seq] | https://www.ncbi.nlm.nih.gov/geo/query/acc.cgi?acc=GSE207611 | NCBI Gene Expression Omnibus, GSE207611 |
| Carraro C, Bonaguro L, Schultze JL, Gatto B | 2022 | Perturbation-informed signatures from crosswise integration of transcriptome and chromatin accessibility analyses predict susceptibility to candidate anticancer drugs (cisplatin validation dataset) [RNA-seq] | https://www.ncbi.nlm.nih.gov/geo/query/acc.cgi?acc=GSE207607 | NCBI Gene Expression Omnibus, GSE207607 |
| Carraro C, Bonaguro L, Schultze JL, Gatto B | 2021 | Perturbation-informed signatures from crosswise integration of transcriptome and chromatin accessibility analyses predict susceptibility to candidate anticancer drugs | https://www.ncbi.nlm.nih.gov/bioproject/PRJNA742032 | NCBI BioProject, PRJNA742032 |

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
