## [Editor Report]

To test differential anticancer drug effects on different tissue types, and to understand drug response mechanism, the authors set up a series of RNA-seq and ATAC-seq experiments on drug responsive and non-responsive cell lines. Then they conducted bioinformatic analyses to pinpoint networks that are altered in responsive vs non-responsive cell lines. Remarkably, they used their analytic results to calculate tumor- and sample-specific response to the drug.

---

## [Decision Letter]

**Decision letter after peer review:**

Thank you for submitting your article "Decoding mechanism of action and susceptibility to drug candidates from integrated transcriptome and chromatin state" for consideration by *eLife*. Your article has been reviewed by 3 peer reviewers, including Murim Choi as Reviewing Editor and Reviewer #1, and the evaluation has been overseen by Mone Zaidi as the Senior Editor. The following individuals involved in the review of your submission have agreed to reveal their identity: Zilu Zhou (Reviewer #2); Jocelynn Pearl (Reviewer #3).

Essential revisions:

The reviewers thought that the quality of the manuscript is high, but also thought that it contains several aspects of potential improvements to assure publication in *eLife*. To briefly summarize their concerns:

1) Design aspect: biological and clinical performance of 3-CePs is not clearly described. Is there any data that illustrates that HCT-15 is low-sensitivity vs BxPC-3 is high-sensitivity?

2) Analytic aspect: is this approach capable of finding signals that are already constitutively expressed in different cell lines? Need more explanation for vCocena and hCoCena, and what are their differences with CoCena2. Were the tests corrected for multiple testing?

3) Application aspect: What about its prediction accuracy within an individual cancer subtype? Will it achieve high prediction accuracy on intra-tumor type heterogeneity? Can the algorithm be applied to other drugs in this same class?

*Reviewer #1 (Recommendations for the authors):*

The crosswise approach is not easily comprehendible. It appears to be a network analysis on the union of genes from RNA-seq and ATAC-seq. I wonder if the introduction can be more clear and state uniqueness compared to existing approaches.

Overall, figure quality is high. However, color coding in figures (cell lines, timepoint, drugs, etc) is unique but not intuitive. Therefore, I had to refer to legends all the way to the last figure. I would suggest changing the color scheme to more intuitively or just use numbers/texts.

*Reviewer #2 (Recommendations for the authors):*

Overall I found this manuscript very interesting to read and significant to the field. However, please consider the following questions/comments:

1. Authors mentioned that HCT-15 is low-sensitivity vs BxPC-3 is high-sensitivity. Is there any data that illustrates this point? Or any description of the phenotype after treatment?

2. Can the authors describe more about vCocena and hCoCena in Method/Supplement, and what are their differences with CoCena2? I wasn't able to find the answer from the provided citation.

3. I am also confused about Figure 5B panel 1 on GFC. I thought GFC is a relative fold change compared to treatment to control. What is the control group in this case to calculate GFC?

4. The pink module is clearly of interest as it showcases the necessity of this multi-omic approach. More generally, can we label all the modules that are only identifiable cross-wise but not by single omic? Can we show a few more detailed examples here? Why are they identified now but not by the single omics approach?

5. The prediction algorithm at the end shows good prediction power across cancer types. What about its prediction accuracy within an individual cancer subtype? Will it achieve high prediction accuracy on intra-tumor type heterogeneity?

*Reviewer #3 (Recommendations for the authors):*

I enjoyed reading your work and the approach you took to integrating multi-omics data to better understand the mechanism of action and epigenetic signatures of 3-CePs in cancer cell lines. The manuscript was clearly written, and I appreciated the thorough results and methods sections which allowed me to clearly follow the majority of the steps taken.

As for major revisions, I believe it would be valuable if the authors would test their sensitivity model for compound B in addition to compound M in cell lines described on page 12 and in Figure 6D.

It is great to see your work in this space and I am hopeful that it further strengthens the use of these data types and analysis methods for improving the drug development process. I will include my general thoughts and suggestions below.

General Thoughts

The paper is long – especially the Results section. But I think for this style of work it is better to have a longer, more thoroughly described results and methods section than to leave out key details.

I would prefer if the paper included p-values for key findings in the text of the article instead of having to locate them in the supplement or figures. I did not see FDRs or q values mentioned throughout the paper/findings – were these calculated? Was there a reason the authors chose to report p-value<0.05 findings as opposed to an FDR/q-value threshold? I find that with gene set enrichment analyses, it can be important to use FDR.

With regards to the crosswise integration method performed, the authors describe a threshold minimum of 15 nodes per cluster on page 23, line 757 of the manuscript. How was this threshold selected? It would be helpful to mention the total sample size input for constructing the network in these sections.

For the sensitivity signature, the authors selected LASSO regression – did the authors consider linear regression? How was LASSO chosen?

For the genes included in the perturbation-informed signature (294 up, 170 genes down), what was the p-value and fold change thresholds used?

For the sensitivity score (described page 12) – do the authors think that this is the ideal model for calculating a sensitivity score? Did they play around with changing the input for the sensitivity model or the model itself? It would be helpful to understand in greater detail why the authors felt that this was the best input for the model.

The validation of the sensitivity prediction on cancer cell lines is given a short, one-sentence description in the results. This could be better explained.

In the discussion and in other locations in the text, the authors describe their approach as 'efficient and versatile' which I think generally ignores the fact that this required next-generation sequencing, two assay types (RNA-seq, ATAC-seq) and in-depth systems biology approaches. Is the argument that the network model is now versatile in that it can be applied to other drugs in this same class? Or that the approach can be borrowed by other groups for other classes of drugs and sample types?

In the methods section, there were a few details I would like to see added. Product numbers including the ATCC product #s, propidium iodide product #, RNA tape station assay. EDTA concentration in ATAC-seq. For clarity, I recommend breaking the RNAseq and ATACseq methods sections apart. Please also include the sample numbers per group that were sequenced. This was hard to find.

---

## [Author Response]

Essential revisions:The reviewers thought that the quality of the manuscript is high, but also thought that it contains several aspects of potential improvements to assure publication in eLife. To briefly summarize their concerns:1) Design aspect: biological and clinical performance of 3-CePs is not clearly described. Is there any data that illustrates that HCT-15 is low-sensitivity vs BxPC-3 is high-sensitivity?

We apologize for not being clear enough when introducing previous findings on the differential sensitivity of HCT-15 and BxPC-3 cancer cell lines to 3-CePs. We now include an introduction to the previous work done by our lab in the revised manuscript (Carraro et al., 2019, Helbing et al., 2020) showing preferential activity of these agents against the pancreatic cancer cell line in both 2D and 3D cancer models (see lines 71-74, 128-129).

2) Analytic aspect: is this approach capable of finding signals that are already constitutively expressed in different cell lines?

We thank the reviewers for pointing out such an interesting concept. The premise for the developed approach is that factors determining the overall cellular sensitivity to a treatment must be determined by basal *intrinsic* characteristics of the cell line. For this reason, we built the sensitivity signature on basal transcriptome profiles, where we prioritized a subset of genes on perturbational evidence (*perturbation-informed basal signature*).

Beyond signature genes, we show in Author response image 1 the results of a GSEA analysis on the whole overlap (300 genes) between DE genes from the baseline comparison (BxPC-3 ctrl vs HCT-15 ctrl) and those from the 6 h M treatment comparison, in the sensitive cell line (BxPC-3 M 6 h vs BxPC-3 ctrl). As expected, pathways like ribosome biogenesis, ROS metabolism, UPR also arise, attesting that genes activated in response to the treatment also have a constitutively different expression in unperturbed cells.

**Author response image 1. sa2fig1:** GSEA analysis on the whole overlap (300 genes) between DE genes from the baseline comparison (BxPC-3 ctrl vs HCT-15 ctrl) and those from the 6 h M treatment comparison, in the sensitive cell line (BxPC-3 M 6 h vs BxPC-3 ctrl). Gene ratio is defined as the count of genes belonging to the specific GO term over the whole DE genes input.

Need more explanation for vCocena and hCoCena, and what are their differences with CoCena2.

We apologize to the reviewers for not having properly described the difference between CoCena and the other two horizontal and vertical approaches. We now included a more detailed description in the Methods section (from line 894) as well as in the main text (lines 393-400).

Briefly, vCoCena and hCoCena are extensions of the original CoCena pipeline for co-expression network analysis aiming at integrating multiple layers of data in one network. hCoCena, first reported in *Aschenbrenner et al., Genome Medicine, 2021*, integrates CoCena networks calculating the correlation between identical features (*genes*) in distinct datasets, making it the ideal tool for integrating different layers of the same omic type. To integrate multi-omics datasets, we further developed and report here vCoCena. A more extended description is provided in Results and Methods sections (from line 894).

Were the tests corrected for multiple testing?

We apologize, if this was not sufficiently described. We specified corrected p values and the applied correction method wherever appropriate throughout the manuscript. To make this more visible, we included a small note in the main text (lines 181, 306, 490, 593), besides having it further specified in the Methods section and figure legends (lines 826, 833, 868, 953, 1042, 1126, 1129, 1135, 1146, 1191, 1197). As specifically asked by Reviewer 3, q values for GSEA on the whole GO dataset are also available in Supplementary files for each biological term. Multiple testing was not applied to FGSEA on gene expression log2FC ranks, being these enrichments inspected on selected GO terms according to the results on the upstream analysis.

3) Application aspect: What about its prediction accuracy within an individual cancer subtype? Will it achieve high prediction accuracy on intra-tumor type heterogeneity?

We thank the reviewers for this interesting question. We now tested the prediction on individual cancer subtypes separately, showing a robust predictive potential (Figure 6 —figure supplement 2 A).

Can the algorithm be applied to other drugs in this same class?

To address this extremely important point, that we agree with the reviewer would be key to prove the versatility of our approach, we further applied the pipeline to the prediction of cancer cell lines’ sensitivity to cisplatin, a thoroughly reported broad-acting chemotherapeutic also acting as a DNA damaging agent. Results strongly supported the broad applicability of our approach, which was able to predict sensitivity to this reference drug with extremely high accuracy.

The inclusion of this other compound allowed us to further improve the prediction pipeline: we realized that a more valuable and balanced vertical network construction could be achieved when equilibrating the input of DE and DAR-associated genes from the RNA-seq and ATAC-seq analyses. This threshold was set to the top 1000 most variable DE genes from RNA-seq and the top 1000 most variable DAR-associated genes from ATAC-seq (instead of all DE and DARs). This allowed to prevent, and this was more evident in the cisplatin dataset, that the ATAC-seq layer could take over the RNAseq one in the final signature construction, just due to the initial much higher number of input entries. Of course, this implied to revise both the CoCena and prediction Figure 5 and 6, with overall analogue findings and an uncompromised prediction goodness. Beyond adjusting the prediction workflow as explained, we also updated the promoter annotation of ATAC-seq peaks to a newer version. This is the reason for the slight changes in Figure 4, which indeed actually improved the overall biological findings.

Reviewer #1 (Recommendations for the authors):The crosswise approach is not easily comprehendible. It appears to be a network analysis on the union of genes from RNA-seq and ATAC-seq. I wonder if the introduction can be more clear and state uniqueness compared to existing approaches.

We apologize to the reviewer for not including sufficient details in describing the difference between the already existing CoCena network approach and the other two approaches, namely horizontal and vertical approaches. We now included a more detailed description not only in the Methods section (from line 894) but also in the main text as suggested by the reviewer (lines 393-400).

Briefly, vCoCena and hCoCena are extensions of the original CoCena pipeline for co-expression network analysis aiming at integrating multiple layers of data in one network. hCoCena, first reported in *Aschenbrenner et al., Genome Medicine, 2021*, integrates CoCena networks calculating the correlation between identical features (*genes*) in distinct datasets, making it the ideal tool for integrating different layers from the same omic technology. To integrate multi-omics datasets, we further developed and report here vCoCena. A more extended description is provided in the Methods section (from line 894).

Overall, figure quality is high. However, color coding in figures (cell lines, timepoint, drugs, etc) is unique but not intuitive. Therefore, I had to refer to legends all the way to the last figure. I would suggest changing the color scheme to more intuitively or just use numbers/texts.

To make the figures reading more intuitive, we included explicit texts besides color coding for each condition in all figure panels where needed (e.g. Figures 1 C, D; 3 A, C; 5 B).

Reviewer #2 (Recommendations for the authors):Overall I found this manuscript very interesting to read and significant to the field. However, please consider the following questions/comments:1. Authors mentioned that HCT-15 is low-sensitivity vs BxPC-3 is high-sensitivity. Is there any data that illustrates this point? Or any description of the phenotype after treatment?

We apologize for not being clear enough when introducing previous findings on the differential sensitivity of HCT-15 and BxPC-3 cancer cell lines to 3-CePs. We now included an introduction to the previous work done by our lab in the revised manuscript showing preferential activity of these agents against the pancreatic cancer cell line in both 2D and 3D cancer models (see lines 71-74, 128-129.

2. Can the authors describe more about vCocena and hCoCena in Method/Supplement, and what are their differences with CoCena2? I wasn't able to find the answer from the provided citation.

We apologize to the reviewer for not having properly described the difference between CoCena and the other two horizontal and vertical approaches. We now included a more detailed description in the Methods section (from line 894) as well as in the main text (lines 393-400).

Briefly, vCoCena and hCoCena are extensions of the original CoCena pipeline for co-expression network analysis aiming at integrating multiple layers of data in one network. hCoCena, first reported in *Aschenbrenner et al., Genome Medicine, 2021*, integrates CoCena networks calculating the correlation between identical features (*genes*) in distinct datasets, making it the ideal tool for integrating different layers of the same omic type. To integrate multi-omics datasets, we further developed and report here vCoCena. A more extended description is provided in Results and Methods sections (from line 894).

3. I am also confused about Figure 5B panel 1 on GFC. I thought GFC is a relative fold change compared to treatment to control. What is the control group in this case to calculate GFC?

Apologies to the reviewer for not being clear enough. GFC is the *Group Fold Change* which was originally described in Aschenbrenner et al., Genome Medicine 2021 as follows: “To assess the expression strength of the found gene clusters in the different studied conditions, the group fold changes (GFCs) of the conditions are calculated for each gene by calculating the mean expression of a gene over all samples and then computing the fold change of the mean gene expression within each condition from the overall mean.” To make this more clear for the reader of the current manuscript, we now included the detailed definition of GFC in the Methods section at line 935.

4. The pink module is clearly of interest as it showcases the necessity of this multi-omic approach. More generally, can we label all the modules that are only identifiable cross-wise but not by single omic? Can we show a few more detailed examples here? Why are they identified now but not by the single omics approach?

We thank the reviewer for this suggestion. In the revised version of the network, the *steelblue* module was reported to illustrate the cross-wise integration adding novel insights into the mechanism of response (lines 451-464). The now so-called *steelblue* module has indeed a similar regulation to the previous *pink* one. We further described in the main text also the other integrated and single-omic modules from vCoCena to further point out the advantage of such network analysis (see lines 427-450).

The overall added value of using vCoCena compared to the initial single layer non-network based analysis consists in the ability to take apart genes with the same regulation pattern, both inside one omic layer and between layers. Grouping transcripts and promoters into the same regulatory module allows to maximize the ratio of hits producing enrichments otherwise not recognizable from separate analysis, due to insufficient representation in the single omics (this is a matter of set size and ratio in this set). Starting both the transcriptome and chromatin horizontal networks from the union of the top 1000 variable genes from both RNA and ATAC-seq datasets, further enhances the multi-omic integration.

5. The prediction algorithm at the end shows good prediction power across cancer types. What about its prediction accuracy within an individual cancer subtype? Will it achieve high prediction accuracy on intra-tumor type heterogeneity?

We thank the reviewer for this interesting question. We now tested the prediction on individual cancer subtypes separately, showing a robust predictive potential (see figure 6 —figure supplement 2 A).

Reviewer #3 (Recommendations for the authors):I enjoyed reading your work and the approach you took to integrating multi-omics data to better understand the mechanism of action and epigenetic signatures of 3-CePs in cancer cell lines. The manuscript was clearly written, and I appreciated the thorough results and methods sections which allowed me to clearly follow the majority of the steps taken.

We thank the reviewer for these very positive statements.

As for major revisions, I believe it would be valuable if the authors would test their sensitivity model for compound B in addition to compound M in cell lines described on page 12 and in Figure 6D.

To further validate the versatility of our platform beyond the specific class of compounds, we further applied the pipeline to the prediction of cancer cell lines’ sensitivity to cisplatin, a thoroughly reported broad-acting chemotherapeutic also acting as a DNA damaging agent. Results strongly supported the broad applicability of our approach, which was able to predict sensitivity to this reference drug with extremely high accuracy.

It is great to see your work in this space and I am hopeful that it further strengthens the use of these data types and analysis methods for improving the drug development process. I will include my general thoughts and suggestions below.General ThoughtsThe paper is long – especially the Results section. But I think for this style of work it is better to have a longer, more thoroughly described results and methods section than to leave out key details.

We thank the reviewer for the positive comment. We also believe that, given the complexity of specific passages in our workflow, it is better to spend a few words more than to take concepts for granted.

I would prefer if the paper included p-values for key findings in the text of the article instead of having to locate them in the supplement or figures. I did not see FDRs or q values mentioned throughout the paper/findings – were these calculated? Was there a reason the authors chose to report p-value<0.05 findings as opposed to an FDR/q-value threshold? I find that with gene set enrichment analyses, it can be important to use FDR.

We apologize with the reviewer, if this was not sufficiently described. We specified corrected p values and the applied correction method wherever appropriate throughout the manuscript. To make this more visible, we included a small note in the main text (lines 181, 306, 490, 593), besides having it further specified in the Methods section and figure legends (lines 826, 833, 868, 953, 1042, 1126, 1129, 1135, 1146, 1191, 1197). q values for GSEA on the whole GO dataset are also available in Supplementary Data tables for each biological term. Multiple testing was not applied to FGSEA on gene expression log2FC ranks, being these enrichments inspected on selected GO terms according to the results on the upstream analysis. We decided to employ GSEA cursorily in different steps of the analysis in order to get a preliminary overview of possible pathways to investigate. The high redundancy observed among significant (p<0.05) terms (see Supplementary file 10) further confirmed the consistency of emerged biological processes.

With regards to the crosswise integration method performed, the authors describe a threshold minimum of 15 nodes per cluster on page 23, line 757 of the manuscript. How was this threshold selected? It would be helpful to mention the total sample size input for constructing the network in these sections.

This is a very valid point. We now report the hit input size upon revising our strategy also in the Result section (line 411) besides having them specified in the Methods section (line 925). Concerning the 15 nodes threshold, we inspected different values from a minimum of 10 and chose such threshold empirically with the overall goal to maximize the number of detected modules with specific regulation patterns (also in horizontally integrated networks) without compromising the possibility to get efficient functional enrichments.

For the sensitivity signature, the authors selected LASSO regression – did the authors consider linear regression? How was LASSO chosen?

We thank the reviewer for this comment. In essence, we agree with the reviewer that probably a simple linear regression without parameter shrinking would also be an option. However, we chose to use l1-regularized linear regression (the lasso), because it is an excellent option in high-dimensional settings. We have recently compared many different approaches in a similar but not identical setting (see Warnat-Herresthal et al., iScience 2020) and could show that LASSO led to very robust results when using transcriptome data in a clinical setting. Based on this experience, we used LASSO. We hope that the reviewer can agree with us that we would not extend the work here to introduce additional other regression models for benchmarking since it would further lengthen the manuscript and would somewhat distract from the main message of the manuscript.

For the genes included in the perturbation-informed signature (294 up, 170 genes down), what was the p-value and fold change thresholds used?

Based on the question of the reviewer and with the new data about Cisplatin integrated into the manuscript we have further improved this section. The M signature was obtained by intersecting genes coming from the selected vCoCena modules with DE genes from the comparison between HCT-15 and BxPC-3 untreated controls using a min abs(log2FC) = 1 and padj < 0.01 to strictly select the most significant hallmark genes. Updated information on the final signature size after revising (both for M and cisplatin) have now been included also in the main text (line 490, 495, 593, 595) besides having it specified in the Methods section (lines 974, 975).

For the sensitivity score (described page 12) – do the authors think that this is the ideal model for calculating a sensitivity score? Did they play around with changing the input for the sensitivity model or the model itself? It would be helpful to understand in greater detail why the authors felt that this was the best input for the model.

This is a very interesting question. Considering the promising predictive outcome, we would argue that the score itself was efficient enough to describe sensitivity. To further support this observation, we now further validated it using a well-established DNA damaging anticancer drug, cisplatin. As we see the experimental validation to be the strongest support for the usefulness of the model, we opted not to further explore the model itself at this time. But we would like to thank the reviewer for this excellent idea and we definitely will continue on the modeling part in a follow-up more computational project to see whether the model itself can be further improved without losing the simplicity of the selected score type. One could also think about extending the model in future developments to scRNA-seq and scATAC-seq data as input. However, we feel that this would be beyond the current scope of the manuscript and we hope the reviewer can agree with this approach.

The validation of the sensitivity prediction on cancer cell lines is given a short, one-sentence description in the results. This could be better explained.

We thank the reviewer for this comment and we addressed this aspect now in the text as suggested at lines 516-521.

In the discussion and in other locations in the text, the authors describe their approach as 'efficient and versatile' which I think generally ignores the fact that this required next-generation sequencing, two assay types (RNA-seq, ATAC-seq) and in-depth systems biology approaches. Is the argument that the network model is now versatile in that it can be applied to other drugs in this same class? Or that the approach can be borrowed by other groups for other classes of drugs and sample types?

Again, these are very valid points by the reviewer. As outlined earlier (see above), we are convinced that – while currently still in its infancy and novel – multi-omics approaches together with in-depth systems biology approaches will revolutionize biology and medicine and will guide the way towards precision medicine (as much as physics was transformed by mathematics in the 20th century, biology and medicine will be transformed by data sciences). However, we agree with the reviewer that we should be more explicit and outspoken. Therefore, to address this extremely important point, and to further illustrate versatility of the approach, we included another well-known drug, cisplatin (as another DNA damaging agent with well-known clinical efficacies), illustrating how efficient our approach predicts cancer cell lines’ sensitivity to cisplatin. We therefore are convinced that the proposed approach is both versatile in being applicable to other drugs and, as a consequence, generalizable to other diseases and target cell populations. We made changes accordingly in lines 695-697.

In the methods section, there were a few details I would like to see added. Product numbers including the ATCC product #s, propidium iodide product #, RNA tape station assay. EDTA concentration in ATAC-seq. For clarity, I recommend breaking the RNAseq and ATACseq methods sections apart. Please also include the sample numbers per group that were sequenced. This was hard to find.

We added these details as suggested by the reviewer in the Methods section (715-717, 750, 797, 799).